# Scale dependence of cirrus heterogeneity effects. Part II: MODIS NIR and SWIR channels

Thomas Fauchez[1,2], Steven Platnick[2], Tamás Várnai[3,2], Kerry Meyer[2], Céline Cornet[4], and Frédéric Szczap[5]

[1]Universities Space Research Association (USRA), Columbia, MD, USA
[2]NASA Goddard Space Flight Center, Greenbelt, MD, USA
[3]University of Maryland Baltimore County: Joint Center for Earth Systems Technology and the Department of Physics , Baltimore, MD, USA
[4]Laboratoire d'Optique Atmosphèrique, UMR 8518, Université Lille 1, Villeneuve d'Ascq, France
[5]Laboratoire de Météorologie Physique, UMR 6016, Université Blaise Pascal, Clermont Ferrand, France

*Correspondence to:* Thomas Fauchez (thomas.j.fauchez@nasa.gov)

**Abstract.** In a context of global climate change, the understanding of the radiative role of clouds is crucial. On average, ice clouds such as cirrus, have a significant positive radiative effect, but under some conditions the effect may be negative. However, many uncertainties remain regarding the role of ice clouds on Earth's radiative budget and in a changing climate. Global satellite observations are particularly well suited to monitor clouds, retrieve their characteristics and infer their radiative impact. To retrieve ice cloud properties (optical thickness and ice crystal effective size), current operational algorithms assume that each pixel of the observed scene is plane-parallel and homogeneous, and that there is no radiative connection between neighboring pixels. Yet, these retrieval assumptions are far from accurate, as real radiative transfer is 3D. This lead to the plane parallel and homogeneous bias (PPHB) plus the independent pixel approximation bias (IPAB) which impacts both the estimation of top of the atmosphere (TOA) radiation and the retrievals. An important factor that determines the impact of these assumptions is the sensor spatial resolution. High spatial resolution pixels can better represent cloud variability (low PPHB), but the radiative path through the cloud can involve many pixels (high IPAB). In contrast, low spatial resolution pixels poorly represent the cloud variability (high PPHB) but the radiation is better contained within the pixel field of view (low IPAB). In addition, the solar and viewing geometry (as well as cloud optical properties) can modulate the magnitude of the PPHB and IPAB. In this Part II of our study, we simulated TOA $0.86~\mu m$ and $2.13~\mu m$ solar reflectances over a cirrus uncinus scene produced by the 3DCLOUD model. Then, 3D radiative transfer simulations are performed with the 3DMCPOL code at spatial resolutions ranging from 50 m to 10 km, for twelve viewing geometries and nine solar geometries. It is found that, for simulated nadir observations taken at resolution higher than 2.5 km, horizontal radiation transport (HRT) dominates biases between 3D and 1D reflectance calculations, but these biases are mitigated by the side illumination and shadowing effects for off-zenith solar geometries. At resolutions coarser than 2.5 km, PPHB dominates. For off-nadir observations at resolutions higher than 2.5 km, the effect that we call THEAB (Tilted and Homogeneous Extinction Approximation Bias) due to the oblique line of sight passing through many cloud columns contributes to a large increase of the reflectances, but 3D radiative effects such as shadowing and side illumination for oblique Sun are also important. At resolutions coarser than 2.5 km, the PPHB is again

the dominant effect. The magnitude and resolution-dependence of PPHB and IPAB is very different for visible, near-infrared, and shortwave infrared channels compared with the thermal infrared channels discussed in Part I of this study. The contrast of 3D radiative effects between solar and thermal infrared channels may be a significant issue for retrieval techniques that simultaneously use radiative measurements across a wide range of solar reflectance and infrared wavelengths.

## 1 Introduction

Clouds cover between 60% to 70% of the Earth's surface and are one of the principal actors in the Earth's radiative budget (Intergovernmental Panel on Climate Change (IPCC) assessment report 5 Boucher et al. (2013)). On average, they lead to a net radiative effect of about $-20\ W \cdot m^{-2}$ (Ramanathan et al., 1989) but this estimation depends on the global circulation model (GCM, Lane et al. (2000); Dufresne and Bony (2008)). It is therefore necessary to better understand clouds and their interaction with radiation. As part of the wide diversity of clouds, high altitude clouds such as cirrus play an important role in the climate and in the Earth's radiation budget (Hartmann and Short (1980); Ohring and Clapp (1980); Liou (1986); Stephens (2005); Eguchi et al. (2007)). Cirrus cover a large part of the Earth's surface (15 % to 40 %, Sassen et al. (2008)) and their high altitude implies a large temperature difference between the cloud top and Earth's surface temperature. Such large difference produces an efficient greenhouse effect by trapping part of the infrared radiation emitted by the surface. Meanwhile, part of the incident solar radiation is reflected to space due to the albedo effect, particularly when the optical thickness is large (greater than 10 (Choi and Ho, 2006)). Most of the cirrus clouds are optically thin (optical thickness less than 3 at 532 nm, Sassen et al. (2008)), leading to an average positive radiative effect (e.g., a greenhouse effect) of about $+28\ W \cdot m^{-2}$ (Boucher et al., 2013). However, their radiative impact depends on numerous factors, such as cloud altitude (Corti and Peter, 2009), cloud thickness (Jensen et al., 1994), crystal shape and size parameter (Min et al., 2010) and temperature (Katagiri et al., 2013). Furthermore, in contrast to the light scattering by spherical water droplets which can be solved using the Mie theory, there is no exact solution for ice crystal scattering due to the multiplicity of crystal sizes and shapes (Lynch et al. (2002)).

Passive satellite sensors are well suited for global temporal and spatial observations of clouds, but the number of retrievable cloud parameters is limited by the information content of the radiative measurements. Cloud optical thickness (COT) and cloud effective particle radius (CER) can be retrieved from space-based radiometric measurements using dedicated operational algorithms. Most algorithms are developed for solar-reflectance bands, like the operational algorithm of the Moderate Resolution Imaging Spectroradiometer (MODIS, Platnick et al. (2017)) for the MOD06 product; Minnis et al. (2011) for the CERES product). Currently, operational constraints such as time constraints or the lack of information regarding the three-dimensional (3D) structure of clouds necessitate the use of a simplified cloud when operationally retrieving cloud properties. In one approach for processing the observations from an area, clouds are considered flat and homogeneous over the entire area. This hypothesis is known as the homogeneous plane parallel approximation (PPHA, Cahalan et al. (1994)). If each cloudy pixel is considered flat, homogeneous and independent of its neighbors, this is called the homogeneous independent pixel approximation (IPA , Cahalan et al. (1994)) or homogeneous independent column approximation (ICA, Stephens et al. (1991)). Such representation

implies that no interaction between pixels or cloudy columns is taken into account between the assumed homogeneous pixels. This is often far from the reality, where clouds have complex three-dimensional and heterogeneous structures and where the radiative transfer occurs in 3D, and this can lead to errors in cloud property retrievals.

Therefore, a means for quantifying the impact of realistic cloud heterogeneities is necessary to begin to understand potential cirrus retrieval errors. Numerous studies have examined this issue for cloud products derived from solar spectral reflectance measurements, but mainly for warm liquid water clouds (e.g. stratocumulus clouds). Indeed, Varnai and Marshak (2001); Zinner and Mayer (2006); Kato and Marshak (2009) and Zhang and Platnick (2011) (and references therein) have shown that neglected cloud horizontal and vertical inhomogeneities can lead to erroneous albedo on top of the atmosphere (TOA)
reflectances estimates, depending on numerous factors such as sensor spatial resolution, the wavelength range, observation geometry, and cloud type, etc. Regarding cirrus clouds in the solar spectral range, Buschmann et al. (2002) found that, for cirrus clouds with mean optical thicknesses smaller than 5 and with relative optical thickness variances smaller than 0.2, retrieval errors due to the cloud homogeneous assumption are smaller than $\pm 10\%$. Carlin et al. (2002) showed that, due to horizontal cirrus inhomogeneity, both solar albedo and outgoing long wave radiation biases could reach $15\ W \cdot m^{-2}$ in magnitude. Using
spectral irradiance measurements below and above tropical cirrus, Schmidt et al. (2009) showed that solar radiation in the visible wavelength range is significantly decreased due to net horizontal radiation transport, especially near cloud edges. Zhang et al. (2010) showed that the vertically homogeneous column assumption used in solar reflectance bi-spectral and thermal infrared retrieval techniques may lead to underestimates of COT and CER of thin cirrus due to the non-linear dependence of ice crystal scattering properties on the effective size. More recently, Fauchez et al. (2012, 2014) showed that 3D thermal infrared
(TIR) brightness temperatures (BT) at TOA can be up to 15 K greater than those computed from a 1D radiative transfer code. (Fauchez et al., 2016) have also showed that, at nadir, 3D radiances are larger than their 1D counterparts for direct emission but smaller for scattered radiation. They have also developed a hybrid model based on exact 3D direct emission, the first scattering order from 1D in each homogenized column, and an empirical adjustment which is linearly dependent on the optical thickness to account for higher scattering orders to drastically reduce the 3D RT computational time. Concerning cirrus cloud optical
property retrievals, Fauchez et al. (2015) showed that cirrus heterogeneity effects can significantly impact them (up to $20\%$ for COT and $100\%$ for CER retrievals) at the 1 km scale of MODIS TIR observations while Zhou et al. (2016) have found similar values for COT retrieved from solar reflectance-based retrievals. In the TIR, TOA BT differences and retrieval errors due to cloud inhomogeneities and 3D effects depend mainly on the standard deviation of optical thickness within a 1 km pixel. At the 1 km scale, the differences between 3D TIR radiative transfer from heterogeneous pixels and 1D radiative transfer from
homogeneous pixels are mainly dominated by the PPHB.

    Most of these previous studies have been performed at the typical 1 km nadir spatial resolution of polar orbiting imagers: however, the impact of the cloud homogeneous assumption depends on the scale at which the cloud is considered homogeneous. For liquid water clouds Davis et al. (1997) have examined cloud heterogeneities as a function of scale for stratocumulus
clouds. They found that the impact of the cloud inhomogeneities on the optical thickness retrieval is at a minimum around 1 km

- 2 km resolution. On the one hand, the 3D radiative impact increases at finer spatial resolutions because the photon mean path becomes as large or larger than the pixel size; conversely, at coarser spatial resolutions, the plane parallel and homogeneous bias (PPHB) is enhanced because the pixel becomes larger than the homogeneity scale. In addition, Zhang et al. (2012) showed that at 100 m spatial resolution, 3D radiative transfer effects, such as side illumination and shadowing, can produce significant differences between CER retrievals based on either 2.1 or 3.7 $\mu m$ reflectances (along with 0.86 $\mu m$) for water cloud. Indeed, the authors showed that 3D effects have stronger impacts on CER retrievals based on 2.1 than on 3.7 $\mu m$, leading to positive difference between the two from cloud side illumination and a negative difference from cloud shadowed side. However, these two opposite effects cancel each other out on the domain average, leading to an overall statistical agreement between the CER retrievals. Yet, at resolutions similar to MODIS, while the two 3D effects cancel each other out, CER retrieved at 2.1 $\mu m$ is systematically larger than the CER retrieved at 3.7 $\mu m$ when averaged over the domain because of cloud horizontal inhomogeneity (PPHB). These results are important for assessing the overall retrieval errors from various space-borne imagers having different spatial resolutions and determining, if possible, which resolution is better to mitigate the effects of cloud heterogeneities on radiance measurements.

In Part I of our study, Fauchez et al. (2017a) discussed the impact of ice cloud (cirrus) heterogeneities as a function of pixel size by simulating MODIS thermal infrared channel measurements. It was shown that the spatial resolution range where the combination of heterogeneity and 3D effects is at its minimum falls between 100 m and 250 m. In Part II of this study, we focus our attention on simulating MODIS visible-near-infrared (NIR) and shortwave infrared (SWIR) reflectance measurements in the 0.86 $\mu m$ and 2.13 $\mu m$ MODIS channels, respectively; these channels are currently used to retrieve cloud optical properties over water surfaces in the operational MODIS cloud product MOD06 (Platnick et al., 2017). The effects of cloud heterogeneity are studied for different viewing and solar angles as a function of spatial resolutions ranging from 50 m to 10 km.

In the next section, we briefly describe the cloud generator model 3DCLOUD (Szczap et al. (2014)) and the 3D radiative transfer model 3DMCPOL (Cornet et al. (2010), Fauchez et al. (2012, 2014)) used to simulate 3D radiative transfer for heterogeneous cirrus clouds. In Section 3 we describe the cloud heterogeneity effects for solar reflectance channels and in section 4, we study the dependence of horizontal heterogeneity effects on spatial resolution and observation geometry. Our summary and conclusions are given in Section 5.

## 2 Simulation of 3D radiative transfer through a realistic 3D heterogeneous cirrus field

The single cirrus field modeled in this paper is identical to the one presented by Fauchez et al. (2017a) in Part I. This allows comparisons of TIR results presented in Part I (Fauchez et al., 2017a) with the NIR/SWIR results in this Part II. The cirrus field is modeled with the 3DCLOUD (Szczap et al. (2014); Alkasem et al. (2017)) code for a mid-latitude summer atmosphere profile (see Fauchez et al. (2017b) Fig. 2) . The scale invariant properties observed for clouds are constrained by 3DCLOUD using a Fourier filtering method to follow a -5/3 spectral slope (Hogan and Kew (2005), Szczap et al. (2014)) . This method

ensures that the horizontal variation of the ice water content (IWC) is consistent with observations (Hogan and Kew (2005); Fauchez et al. (2014)).

The radiative transfer simulations are then performed using the 3DMCPOL Monte Carlo radiative transfer (RT) code (Cornet et al. (2010), Fauchez et al. (2014)) assuming cyclic boundary conditions are imposed at the edges of the domain. Cirrus optical properties are parameterized using the same microphysical model assumed by the MODIS Collection 6 (MOD06) cloud product, namely the severely roughened single-habit column aggregate from Yang et al. (2013). The domain mean optical thickness of the 3DCLOUD cirrus is 1.5 at $0.86~\mu m$, and the cloud is assumed to have a constant CER of $10~\mu m$. Note that while the microphysical properties are homogeneous, the extinction coefficient varies horizontally and vertically. The optical properties of the assumed ice crystals used at two MODIS channels are shown in Table 1.

Solar reflectances are computed with 3DMCPOL for MODIS channel 2 (centered at $0.86~\mu m$) and 7 (centered at $2.13~\mu m$). Five solar geometries are considered: $(\Theta_s = 0°; \Phi_s = 0°)$, $(\Theta_s = 30°; \Phi_s = 0°, 90°$ and $180°)$ and $(\Theta_s = 60°; \Phi_s = 0°)$ with $\Theta_s$ and $\Phi_s$ corresponding to solar zenith and azimuth angles, respectively. For each of these solar directions, reflectances are computed for twelve viewing geometries: three viewing zenith angles ($\Theta_v = 0°, 30°$ and $60°$) and four viewing azimuth angles ($\Phi_v = 0°, 45°, 90°$ and $180°$). The azimuths of viewing angles relative to the cirrus field are represented in Figure 1. This figure shows the cirrus cloud field simulated by 3DCLOUD based on the mid latitude summer meteorological profiles already used in Fauchez et al. (2017b). No aerosol is added and the surface is Lambertian with a constant albedo of 0.05. Top panels show the vertical profiles of ice water content (IWC) along the red lines in the optical thickness field, which is shown in the bottom panels for different viewing angles. Note that the cirrus vertical profile looks very different along different azimuths. The wind direction is following the $\Phi_v = 45°$ arrow leading to clearly visible virga features at this angle. Note that the azimuth view angles $\Phi_v = 225°$ and $270°$ are shown on this figure, but reflectances are not computed for these viewing azimuths due to computational time limitations.

Figure 2 (a) shows the cirrus optical thickness field at $0.86~\mu m$ at 50 m spatial resolution, and Fig. 2 (b) shows the corresponding 3D reflectance field at nadir. One hundred billions of fictive light particles (FLIPs (Pujol, 2015), referenced hereafter as photons) are computed in 3.5 days on 2048 parallel cores of the NCCS discover supercomputer (see acknowledgements) for 3D computations of solar reflectances within an accuracy of about $0.1\%$.

## 3    Decomposition of the effects of cloud heterogeneity and 3D radiative transfer on simulated reflectances

Clouds are complex 3D structures where solar and terrestrial radiations propagate in a three-dimensional space. However, in current retrieval algorithms, for simplification and/or computational reasons, the homogeneous independent pixel approximation (IPA, Cahalan et al. (1994)) is commonly applied: each portion of the observed cloudy scene is sampled in pixels, and each

pixel is assumed to be horizontally homogeneous as well as radiatively independent of its neighbors (1D radiative transfer assumption). The sub-pixel horizontal heterogeneity leads to the plane-parallel and homogeneous bias (PPHB) because of the non-linearity between optical properties and radiance/reflectance. The 1D assumption leads to several effects describing below in terms of 3D radiative effects. Both effects (IPA bias and PPHB) are strongly dependent on the sensor spatial resolution. The sub-pixel heterogeneity effects increase for coarser spatial resolutions, while 3D effects linked to net horizontal photon transport between columns increase for finer spatial resolutions. The range of spatial resolutions for which either the IPA biases or the PPHB are dominant depends on the wavelength. Off course for thermal wavelength no illumination and shadowing effects are present and in addition cloud absorption is much larger for thermal infrared than for solar wavelengths leading to larger PPHB but smaller IPA effect (because of less scattering).

To study cirrus heterogeneities that can affect solar reflectances, we simulate 1D solar reflectances with 3DMCPOL following the IPA assumption. COT is first averaged from the highest spatial resolution (50 m) to the spatial resolution of interest (up to 10 km) before performing the 1D RT calculations. In turn, 3D reflectances are always computed at 50 m resolution and are then aggregated to a given spatial resolution (from 50 m to 10 km). The difference $\Delta R$ between 3D and 1D reflectances obtained this way corresponds to the total bias including sub-pixel cloud heterogeneities and IPA biases.

To go further, it is useful to distinguish the amplitude and scale of the different effects and radiative processes that impact cloud-top reflectances when the homogeneous and IPA assumption are used:

1. The Plane-Parallel and Homogeneous Bias assumption (PPHB);

2. Radiative impact of vertical inhomogeneity (not considered here);

3. The Tilted and Homogeneous Extinction Approximation Bias (THEAB); and

4. 3D radiative effects due to the radiative non-independence of the pixels.

**Plane-parallel and homogeneous assumption bias (PPHB):** In current operational satellite retrieval algorithms, the scene within each observed pixel is assumed to be horizontally homogeneous. The impact of the sub-pixel horizontal heterogeneity clearly depends on the sensor spatial resolution (Oreopoulos and Davis, 1998). In Fig. 3 we plot the reflectances $R_{50m}^{1D}$ estimated with a 1D RT calculation at 50 m for $0.86 \ \mu m$ and $2.13 \ \mu m$ channels, respectively, as a function of the 50 m optical thickness at $0.86 \ \mu m$. We see that the relation between reflectance and optical thickness is increasing (the thicker the cloud, the more light reflected) but is non-linear. Indeed, the reflectance of the averaged optical thickness $R_{\overline{COT}}$ is larger than the average reflectance of the optical thicknesses $\overline{R1D}$. This is Jensen's inequality (Newman et al., 1995), usually called the plane-parallel and homogeneous bias (PPHB, Cahalan et al. (1994); Cahalan et al. (1995), Oreopoulos and Davis (1998)). Note that we plot 1D but not 3D reflectances specifically to highlight the effect of the PPHB.

**Vertical inhomogeneity (not considered here):** For the same optical thickness, the vertical distribution of the ice crystal CER and IWC may have an impact on TOA reflectances and cloud retrievals. For instance, Zhang et al. (2010) showed that the

vertically homogeneous column assumption used in solar reflectance bi-spectral and thermal infrared retrieval techniques may lead to underestimates of COT and CER of thin cirrus due to the non-linear dependence of ice crystal scattering properties on the effective particle size. However, since we are interested in the impact of the space sensor horizontal spatial resolution on TOA NIR/SWIR reflectances, we do not consider the vertical heterogeneity.

**Tilted and homogeneous extinction approximation bias (THEAB):** This effect concerns off-nadir viewing geometries. At the spatial resolution of a spaceborne imager, the tilted line of sight cross several atmospheric columns above the surface for a single observed cirrus pixel, while in the operational algorithms, each cloudy column is assumed horizontally infinite and homogeneous. More detailed explanation will be given on section 4.3 but this effect can lead to a smoothing of the radiative field from an sideview because each line of sight encounters many cloudy columns and voxels (pixel in volume) of various optical properties. This is shown in Fig. 4 where we can see the oblique line of sight crossing voxels of various extinction while the IPA considers only the column underneath the observe pixel. The THEAB column, which is then uprighted, is therefore different from the IPA column. Note that both column has the same vertical extension, but for the THEAB the extinction in each voxel has been adjusted to account for the longer oblique path. Statistically, each line of sight will cross voxels with a wide range of extinction along its path leading to similar total optical paths between each of these line of sight. (e.g. Várnai and Davies (1999); Varnai and Marshak (2003), Kato and Marshak (2009)).

**3D radiative effects:** In addition to the impact of the heterogeneity in the cloudy column, the IPA can lead to significant retrieval errors due to the horizontal photon transport between nearby columns (Várnai and Davies (1999); Varnai and Marshak (2001, 2003); Marshak and Davis (2005), Oreopoulos and Cahalan (2005), Kato and Marshak (2009), etc.). Indeed, for 3D radiative transfer, photons can cross several cloudy columns having different optical properties, though horizontal transport depends strongly on particle absorption and can vary widely between NIR and SWIR imager channels for the same pixel (e.g., Platnick (2001)). Three distinct categories of 3D effects are worth mentioning:

1. **Horizontal radiation transport (HRT)** (Davies (1978), Kobayashi (1993),Davis and Marshak (2001), Várnai and Davies (1999); Varnai and Marshak (2003)): photons can be transported from one cloud column to another. Marshak et al. (1995) determined that the radiative smoothing scale $L$ due to photon horizontal transport (or photon diffusion) is expressed by $L = H \times \sqrt{(1-g)COT}$ where $H$ is the cloud geometrical thickness, COT the optical thickness and "$g$" the asymmetry parameter of the phase function. Horizontal transport leads to the photon's escapes from the cloud (leakage effect) where the optical thickness is the thinnest because first, photons in thin columns have less chance to be absorbed or scattered toward another column, and second, because photons in neighboring columns with stronger scattering have a higher chance to leave the cloud if they are scattered toward a neighboring column with a smaller extinction coefficient. Therefore, the net flux of photons tends to flow from thick to thin regions. This is illustrated in Fig. 5 where are plotted 3D and 1D Nadir reflectances at 1 km as a function of the optical thickness and where we can see that the 1 km reflectances are smaller in 3D than in 1D for large COT while the opposite is true for small COT. This increases

the reflectance of optically thin pixels and decreases the reflectance of optically thick ones. This is confirmed by Fig. 6 (a) that shows nadir reflectances at 50 m ($R_{50m}$) as a function the optical thickness $\tau^{0.86\ \mu m}$ in 3D and 1D for channels centered at 0.86 $\mu m$ and 2.13 $\mu m$ for a solar zenith angle of $\Theta_s = 0°$. We note, however, that the impact of density variations on the flow of net radiation is inherently scale dependent. For example, large-scale structures may guide the net radiation to flow toward and through small pockets of high density. For zenith sun we can see that, due to the HRT, for small $\tau^{0.86\ \mu m}$ ($\leq 2$), 3D reflectances are larger than 1D reflectances while for larger $\tau^{0.86\ \mu m}$, 3D reflectances are smaller than 1D reflectances. We note that HRT, as described above, dominates only for not too tilted sun ($\Theta_s = 0°$ for Fig 6a and $\Theta_s = 30°$ for 6b). For oblique sun (Fig 6c), the trend reverses as 3D reflectances exceed 1D ones for optical thicknesses larger than about 5 and 3D reflectances are lower than 1D ones for smaller optical thicknesses. Increase of 3D reflectances oblique sun is caused by the side illumination discussed below.

2. **Side illumination effect** (Wendling (1977), Varnai and Marshak (2003), Zhang et al. (2012)): This effect occurs when photons of the incoming sunlight travel obliquely and enter a cloud through its side, rather than the top. In contrast to the HRT, side illumination tends to increase the reflectance of thicker clouds for backward and overhead viewing directions (Loeb and Davies, 1996) as we can see in Fig. 6 (c), where most of the 3D reflectances are larger than 1D reflectances.

3. **Shadowing effect** (L. H. Chambers (1997), Zuidema and Evans (1998), Varnai and Marshak (2003), Zhang et al. (2012)): Similarly to side illumination, this effect occurs when incoming solar photons traveled diagonally, but this time, photons first reach a cloudy column with a large extinction, which blocks incoming photons from reaching the thinner columns behind it ("upward trapping" process illustrated in Fig 5a of Varnai and Davies (1999)).

The first 3D effect acts to smooth the radiation field structure, whereas the second and third effects lead to a roughening effect of the radiation field. Smoothing is an isotropic effect accounting for large scattering orders, whereas roughening occurs mainly in the solar plane by affecting direct and low order scattered sunlight (Zuidema and Evans (1998); Varnai and Marshak (2003); Oreopoulos and Cahalan (2005)). Also, the side illumination effect is usually larger than the shadowing effect (Varnai and Marshak (2003)).

Note that all of these effects are dependent on the cloud optical thickness heterogeneity, the vertical inhomogeneity of the volume extinction, the variation of the cloud top and base altitude (always considered flat in our study) as well as the solar and viewing angles. The total effect due to cloud inhomogeneity and 3D radiative transfer is therefore very complex and dependent on the spatial resolution.

## 4 Cirrus horizontal inhomogeneity and 3D effects as a function of the observation scale

### 4.1 Overall differences between 3D and 1D reflectances

In nature, radiative transfer occurs in 3D not in 1D. 3D radiative effects influence the spectral reflectance of a given pixel due to its radiative connection to its neighbors. These 3D effects includes the HRT between cloudy columns or side illumination and shadowing effect for oblique Sun illumination. To compare reflectances from a 3D radiative transfer through a heterogeneous pixel with reflectances from the 1D homogeneous pixel assumption, we estimated the arithmetic mean difference between aggregated 3D at x km and 1D reflectancesof the mean optical thickness at x km, with respect to the 3D aggregated reflectance in percentage, as follows:

$$\overline{\Delta R}(\overline{3D} - 1D)\ (\%) = \frac{100}{\overline{R^{3Dxkm}}} \times [\sum_{i=1}^{N}(\overline{R^{3D}}^{xkm} - R^{1Dxkm})]/N, \tag{1}$$

where $\overline{R^{3D}}^{xkm}$ is the averaged of 3D reflectances computed at 50 m resolution, $R^{1Dxkm}$ is the 1D reflectances computed for the optical thickness averaged over $x\ km$ and $N$ is the number of pixels at the spatial resolution $x\ km$. Note that the comparison here shows the total bias including how the nonlinearity of the relationship between reflectance and optical thickness, and the 3D radiative effects, affects TOA reflectances for a given view angle and spatial resolution.

Some effects such as the HRT may have almost no impact on average reflectances but locally, at the pixel scale, they may have large positive and negative effects. We therefore estimate the mean absolute magnitude of the total effect by calculating the absolute mean difference between aggregated 3D and 1D reflectances, relative to the 3D aggregated reflectance in percentage, as follows:

$$\overline{|\Delta R(\overline{3D} - 1D)|}\ (\%) = \frac{100}{\overline{R^{3Dxkm}}} \times [\sum_{i=1}^{N}(|\overline{R^{3D}}^{xkm} - R^{1Dxkm}|)]/N, \tag{2}$$

Figure 8 shows $\overline{\Delta R}(\overline{3D} - 1D)$ (panels (a), (b) and (c)) and $\overline{|\Delta R(\overline{3D} - 1D)|}$ (panels (d), (e) and (f)) at $0.86\ \mu m$ as a function of the spatial resolution (ranging from 50 m to 10 km), for various viewing and solar angles. First, we see that $\overline{\Delta R}(\overline{3D} - 1D)$ is negative for most of the spatial resolutions, viewing and solar angles and is larger than the 3% MODIS reflectance measurement uncertainty. $\overline{\Delta R}(\overline{3D} - 1D)$ and especially $\overline{|\Delta R(\overline{3D} - 1D)|}$ tend to be the smallest for nadir view, and they are almost constant over wide ranges of spatial resolutions and $\Theta_s$. For oblique views, the larger the viewing zenith angle $\Theta_v$, the larger is $\overline{\Delta R}(\overline{3D} - 1D)$ and $\overline{|\Delta R(\overline{3D} - 1D)|}$, except for $\Phi_v = 45°$. This view is directly parallel to the fallstreaks of the cirrus, where the variability along the line of sight is the smallest (see Fig. 1 (b) and (e)). We can also see that $\overline{\Delta R}(\overline{3D} - 1D)$ is positive for $\Theta_v = 60°$ for several $\Phi_v$ for the finest spatial resolutions (see section 4.3 on the THEAB).

Figure 9 is the same as Fig. 8 but for 2.13 $\mu m$ reflectances. We can see that the amplitude of $\overline{\Delta R(3D-1D)}$ and $\overline{|\Delta R(3D-1D)|}$ are very similar. Because of this similarity between the effects on NIR and SWIR reflectances, in the later figures we only focus on the NIR channel centered $0.86\ \mu m$ to avoid overloading the manuscript.

5    In Fig. 10, we can see the influence of the solar azimuth angle on $\overline{\Delta R}$ and $\overline{|\Delta R|}$. The weakest absolute effect $\overline{|\Delta R|}$ is for a solar azimuth angle $\Phi_s = 0°$. However, the differences are relatively small over the solar azimuth angles because each of the three angles highlights the cirrus fallstreaks obliquely (see Fig. 1 to compare with viewing angles). Indeed, while cirrus clouds with fallstreaks are particularly heterogeneous as illuminated from different solar azimuth angles, the relative small optical thickness of cirrus does not lead to large azimuthal dependency.

## 4.2    Plane parallel and homogeneous bias

When all the various effects relative to pixel optical property inhomogeneity, radiation transport, and oblique viewing geometry act together, it is difficult to separate their relative contributions. Following Varnai and Marshak (2003), the horizontal inhomogeneity effects due to the PPHB can be isolated from 3D effects by using 1D radiative transfer calculations. Nadir
1D reflectances aggregated from the native spatial resolution (50 m) can be compared to reflectances computed at a given spatial resolution following the homogeneous pixel assumption. This difference, relative to the 3D aggregated reflectance in percentage, is expressed by Eq. 3, and is shown in Fig. 11.

$$\overline{\Delta R(\overline{1D}-1D)}\ (\%) = \frac{100}{\overline{R^{3Dxkm}}} \times [\sum_{i=1}^{N}(\overline{R^{1D}}^{xkm} - R^{1Dxkm})]/N, \tag{3}$$

where $\overline{R^{1D}}^{xkm}$ and $\overline{R^{3D}}^{xkm}$ denote the averaged 1D and 3D reflectances, respectively, computed at 50 m and $R^{1Dxkm}$ is the
1D radiance computed for the averaged optical thickness of $x\ km$-size areas.

As we can expect $\overline{\Delta R(\overline{1D}-1D)}$ is negative, the PPHB increases overall as the pixel size increases with the largest $\overline{\Delta R(\overline{1D}-1D)}$ occurring at 10 km spatial resolution, i.e. when the entire cloud field is assumed homogeneous. As we have already seen in Fig. 3, in most of the optical thickness range ($\geq 2$), the PPHB leads to averaged reflectances smaller than the reflectances computed for the averaged optical thickness, such that, on average, $\overline{\Delta R(\overline{1D}-1D)} < 0$. The PPHB is the dominant
effect at coarse spatial resolutions, explaining while the total bias $\overline{\Delta R(3D-1D)}$ in Fig. 8 is negative for every viewing and solar angles at resolutions coarser than 1 km. Also, we can see that for an overhead sun (panel (a)), the PPHB $\overline{\Delta R}$ is smaller than the MODIS reflectance measurement uncertainty of $\pm 3\%$ (Xiong et al., (2005, 2017)) represented by the horizontal dash lines, except for $\Theta_v = 60°$ (blue lines) from 500 m spatial resolution and beyond. Therefore, for this particular scene and an overhead sun, the PPHB is not significant except for very large viewing angles. Also, we can see that the PPHB increases with
increasing solar zenith angle (moving from (a) to (c)) because of the increasing of the non-linearity of the reflectance vs. COT relation (compare the blue or magenta curves of 1D reflectances in Fig. 6 (a) and (c)) making the PPHB (which arises from

this nonlinearity) stronger. But again, the PPHB becomes significant for spatial resolutions coarser than 250 m for a very large viewing zenith angle ($\Theta_v = 60°$) at low solar zenith angle, or at very high solar zenith angle ($\Theta_s = 60°$) at any viewing angle. This conclusion is different from that of Fauchez et al. (2017a) for thermal infrared wavelengths, where the PPHB dominates for Nadir view beyond $\sim 250 \, m$. Indeed in the TIR, the cloud absorption is larger and the source of radiative emission is not the Sun but the atmosphere, the cloud and the surface. Furthermore, the large temperature difference between the cirrus and the surface leads to large brightness temperature inhomogeneities and therefore large PPHB.

### 4.3 Tilted and homogeneous extinction approximation bias (THEAB)

When the viewing zenith angle is large, the bias due to the tilted view of the cloudy scenes called THEAB may also significantly impact the difference between TOA reflectances estimated with the 1D horizontal homogeneous cloud assumption and those corresponding to the reality of 3D radiative transfer. For cloud observations from TOA, an oblique line of sight may cross many different cloudy columns, while the 1D plane parallel and homogeneous assumption considers only a single cloudy column above the observation pixel, assumed horizontally infinite with vertically heterogeneous extinction coefficient (see Fig. 4). In essence, the Tilted and Homogeneous Extinction Approximation (THEA) can be considered a variant of the Tilted Independent Pixel Approximation (TIPA) used in earlier studies (e.g., Várnai and Davies (1999); Wapler and Mayer (2008); Frame et al. (2009), Wissmeier et al. (2013)), but with the tilting based on the view direction instead of the solar direction. A somewhat similar concept to THEA was used in Evans et al. (2008), where reflectances were related to cloud properties calculated along the slanted line of sight. Each tilted line of sight crosses large, medium and small extinctions through many different columns leading to an average optical paths similar between each tilted columns and therefore the field of view appears more homogeneous than the one with independent cloudy columns (1D assumption) with small optical thickness juxtaposed to large optical thickness. This effect is shown in Fig. 12 where we can see that the optical thickness field at 50 m spatial resolution view from $60°$ zenith angle is much smoother than the one see from nadir. We can also see that the extinction plumes are stretched out, spreading and smoothing the cloud extinction over the columns. Indeed, for a voxel horizontal and vertical sizes of 50 m and 72 m, respectively, and a $\Theta_v = 60°$, the line of sight reaching the top of a given voxel from its center (see Fig. 4) then cross horizontally $72 \times tan(60) \sim 125 \, m$, i.e. two adjacent voxels before reaching the underneath cloud layer. To highlight only the THEAB without considering the horizontal radiative transport effect we compared 1D reflectances computed with the homogeneous, independent and infinite pixel assumption (named $1D$) to those computed with the independent but non-infinite pixel assumption (named $1D_{o.e}$ for oblique extinction). In the latter situation, the line of sight cross neighboring cloudy columns, but the columns were still radiatively independent (i.e., this is not 3D RT). For each pixel, we re-created a 1D cloud for which the optical thickness per layer corresponds to the oblique optical thickness of the 3D heterogeneous extinction field but keeping the cell dimension constant. We have ran 1D RT using the oblique columns crossed as adjacent vertical cloud layers (i.e., tilted the oblique columns crossed to a vertical column). The relative and absolute differences, with respect to the

3D aggregated reflectance in percentage, are estimated following equations 4 and 5, respectively, and are also represented in Fig. 13 (a), and (b), respectively.

$$\overline{\Delta R}(1D_{o.e.} - 1D)\ (\%) = \frac{100}{R^{3Dxkm}} \times [\sum_{i=1}^{N}(R_{o.e.}^{1Dxkm} - R^{1Dxkm})]/N, \tag{4}$$

$$\overline{|\Delta R}(1D_{o.e.} - 1D)|\ (\%) = \frac{100}{R^{3Dxkm}} \times [\sum_{i=1}^{N}(|R_{o.e.}^{1Dxkm} - R^{1Dxkm}|)]/N, \tag{5}$$

In Fig. 13, we see that for a viewing zenith angle of $\Theta_v = 30°$, the relative value $\overline{\Delta R}$ is, on average, equal or below the MODIS reflectance measurement uncertainty, while locally the absolute value of THEAB ($\overline{|\Delta R|}$) can reach few tens of percents. The THEAB is more important for large viewing zenith angles because the number of columns crossed by the line of sight increases with the viewing zenith angle. The THEAB also increases with the solar zenith angle from $30°$ to $60°$ for much of the same reasons. There is a small dependence on the viewing azimuth angle with the largest effect at $\phi_v = 180°$.

Because the optical thickness field from the $1D_{o.e.}$ appears more homogeneous than for 1D for the finest spatial resolutions, the difference $R_{o.e.}^{1Dxkm} - R^{1Dxkm}$ is positive. This explain why for the finest spatial resolution and viewing zenith angle the total bias $\overline{\Delta R}(3D - 1D)$ in Fig. 8 (0.86 $\mu m$) and Fig. 9 (2.13 $\mu m$) is positive. By comparing these results with those of Fig. 8 we can see that, for spatial resolutions below 1 km, the THEAB is the dominant effect for large solar zenith angles. The absolute THEAB effect $\overline{|\Delta R}(1Do.e - 1D)|$ is even larger than the total effect $\overline{|\Delta R}(3D - 1D)|$ which is reduced by the

radiative smoothing. At a pixel size of 2.5 km, the large pixel size reduces the THEAB, which becomes almost zero, as when observation pixel resolution increases, differences between tilted and vertical optical thicknesses logically reduce. ; Thus, the PPHB thus becomes the dominant effect. Note that we choice to calculated the THEAB instead of the TIPA bias because only the former helps to understand why $\overline{\Delta R}$ is positive for the small scales and negative for the larges, even when the Sun is at zenith (no TIPA bias). The TIPA bias is implicitly included in the 3D effects discussed in section 4

**4.4   3D effects**

Like other effects, the importance of 3D effects is dependent on the spatial resolution. 3D effects refer to both radiative (HRT, side illumination and shadowing) and geometric (THEAB) 3D effects. In Fig. 14 we can see the relative difference due to 3D effects calculating from the total difference minus the PPHB (in percentage relative to the 3D reflectances) for a Sun at zenith and for viewing angles of $\Theta_s = 0°$, $30°$ and $60°$. The difference, averaged over the field, for each spatial resolution is

25 represented by the solid line and the 10th and 90th percentiles of the difference are represented by the shaded area. Note that Fig. 14 is for sun at zenith, consequently only HRT are present with no shadowing and illumination effects. We can see that, at the pixel scale, HRT can have a positive or negative impact on the reflectance but that the average value (averaged over the entire field) is constant over the spatial resolutions. Also, as expected, we see that the amplitude of HRT is much larger for fine (up to $+150\,\%$, down to $-200\,\%$) and decrease with the decrease of the spatial resolutions. Note that negative differences are

larger than positive differences. Negative reflectance differences are associated with a darkening of the radiation because the HRT reduces the reflectances of large optical thicknesses (see Fig. 5). To not overload Fig. 14 with too many viewing and solar angle we have summarized the relative 3D effects averaged over the 10 km field in Table 2. The interpretation of Table 2 is not easy and further study need to be conducted to give consolidated conclusions but some clues can still be given. In this table, we can see that for a Sun at zenith (no side illumination nor shadowing effects) and a view from nadir (no THEAB), 3D effects correspond to HRT only and the averaged difference is negative. However, for a Sun at Zenith (first line) and for the largest viewing zenith angles $\Theta_v = 60°$, the differences are less negatives and are even positives for viewing azimuths $\Phi_v = 90°$ and $\Phi_v = 180°$. This is due to the THEAB which increase the reflectances (see Fig. 13). This effect is less strong at $\Phi_v = 0°$ and is weak at $\Phi_v = 45°$ because the line of sight is parallel to the fallstreaks leading to i) a similar extinction through the crossed columns and ii) a reducing of the fallkstreaks reflectances due to HRT. For a Sun off-zenith, the effects of shadowing and side illumination are added to those of the HRT. At $\Theta_s = 30°$, we can see that the negatives differences dominate because of the HRT, except again for for $\Theta_v = 60°$ with $\Phi_v = 0$, 90 and 180°. For $\Theta_s = 60°$, two effects can reduced the negative differences and even lead to positive differences. The illumination effects that increase the 3D reflectances and the THEAB, in particular for the $\Theta_v = 60°$.

## 5    Conclusions

In this work, we have modeled a typical cirrus cloud field, with a constant $CER = 20$ $\mu m$ using the 3DCLOUD model . Only one cirrus structure has been modeled for computational time reasons, but many spatial resolutions, viewing and solar angles have been considered. However, the radiative processes discussed here can be generalized to other cirrus clouds, with differences depending on cirrus structure (whether fallstreaks are included or excluded), solar and view geometries, average optical thickness etc. 0.86 and 2.13 $\mu m$ reflectances have been simulated for this scene using the 3DMCPOL code for various configurations: 1) Full 3D radiative transfer at high spatial resolution, 2) 1D radiative transfer at various spatial resolutions, taking into account the extinction variability along the oblique line of sight, and 3) Vertically homogeneous 1D radiative transfer at various spatial resolutions. The spatial resolutions considered here range from 50 m to 10 km. By comparing the results of these simulations, the paper examined three types of effects: the plane-parallel and homogeneous approximation bias (PPHB) due to the non-linear relationship between optical thickness and reflectance, the tilted and homogeneous extinction approximation bias (THEAB) that arises because in 1D, the line of sight is assumed to remain in a single vertical column, while in reality, it can cross many different cloudy columns and the 3D effects due to 3D radiative effects such as the horizontal transport of photons between pixels (HRT) or (for oblique sun) side illumination and shadowing effects, associated with the 3D geometrical effect of the THEAB. The relative contribution of these three effects to the TOA reflectances is strongly dependent on spatial resolution but also on cloud structure. No particular differences have been noticed between 0.86 and 2.13 $\mu m$ channels (except in the magnitude of the effects); therefore, for clarity, most of the figures show results for the 0.86 $\mu m$ channel only. For the particular configuration of a cirrus uncinus, we have emphasized the following points:

– For nadir observations:

- Below 2.5 km spatial resolution, 3D effects are dominant.

    • For overhead Sun, HRT is the only 3D effect and can reach $+20$ % and $-60$ % in a 50 m spatial resolution pixel.

    • For oblique Sun, 3D radiative effects such as side illumination and shadowing modify the differences between 3D and 1D reflectances which can reach $+120$ % and $-170$ % ($\Theta_s = 30°$) and $+150$ % and $-200$ % ($\Theta_s = 60°$) in a 50 m spatial resolution pixel.

- At spatial resolution coarser than 2.5 km, the PPHB is the dominant effect.

  - For off-nadir observations:

    - Similar conclusion to nadir observations except that below 2.5 km spatial resolution and for a very large viewing zenith angle of $\Theta_v = 60°$ and an viewing azimuth not parallel to the fallstreaks, In addition to illumination effects, the THEAB leads to increase the 3D reflectances (radiative) effects.

For off-nadir observations, the THEAB is large mostly for $\Theta_s = 60°$ and for high spatial resolutions (small pixels, roughly below about 1 km), especially for a line of sight crossing perpendicularly the fallstreaks of cirrus uncinus. This bias is difficult to evaluate from observations, as this would need an active sensor, such as a lidar, looking at an oblique view angle. For low spatial resolutions (large pixel sizes, roughly > 2.5 km) the PPHB is the largest effect when compared to higher spatial resolutions. Note that this spatial resolution is slightly larger than the roughly > 1 km estimated by Davis et al. (1994) for stratocumulus clouds; the difference is certainly due to the weaker optical thickness of the studied cirrus cloud. Between these two ranges, competition between 3D effects, THEAB and PPHB are complicated and will depend on the cloud structure and the viewing and solar geometries. It is therefore difficult to generalize the conclusions for this intermediate range to other cirrus clouds. Concerning 3D effects, the relative influence of the HRT (leading to a net flux of photons mostly from thick to thin regions), versus side illumination and shadowing effects which (mostly increasing the reflectance of large optical thicknesses, but blocking the photons from eventually reaching thinner neighbor regions) is dependent on the Sun zenith angle. At moderate Sun zenith angles ($0°$ and $30°$ in our simulations), the HRT is dominant over the side illumination effects leading overall to a negative 3D effect. However when the Sun is very low ($60°$ in our simulations), the side illumination mitigate the effects of both, HRT and shadowing leading to weaker 3D effects. Note that, 3D effects, when averaged over the entire field, are constant whatever the spatial resolutions. Overall, the total difference between 3D and 1D reflectances increases with the solar zenith angle because of the increase of PPHB with the solar zenith angle as already shown by Loeb and Davies (1996) on stratocumulus clouds. The overall predominant effect will therefore depend on the cloud optical thickness and viewing/solar geometries. Note that the results do not significantly change with a larger CER than $20\mu m$ for $0.86$ $\mu m$ because the optical properties are fairly constant up to CER of $50$ $\mu m$, but at $2.13$ $\mu m$ the absorption increases with CER leading to stronger PPHB and weaker 3D effects (because the mean free path is reduced by the absorption).

In Part I of this study, which focused on the impact of cloud inhomogeneity and 3D effects in thermal infrared channels, the PPHB was found to be larger than 3D effects at resolutions coarser than 100-250 m. This is because the cloud absorption is

much larger in the thermal infrared (TIR), leading to a larger PPHB even at relatively small pixel sizes. The differences between horizontal inhomogeneity and 3D effects at TIR and NIR/SWIR channels pose a problem for retrieval techniques such as the optimal estimation method (OEM, Rodgers (2000)) that use multiple channels from these wavelength ranges (Fauchez et al., 2017b). In a future study, the impact of such differences in the retrieval of cirrus optical properties will be investigated using an OEM at five channels across the NIR/SWIR/TIR ranges and at spatial resolutions ranging from 50 m to 10 km.

*Code availability.* The simulated data used in this study were generated by the 3DCLOUD (Szczap et al. (2014)) and 3DMCPOL (Cornet et al. (2010); Fauchez et al. (2014)) closed-source codes. Please contact the authors for more information.

*Acknowledgements.* The authors acknowledge the Universities Space Research Association (USRA) through the NASA Postdoctoral Program (NPP) for their financial support.

We thank the UMBC High Performance Computing Facility (HPCF) for the use of their computational resources (MAYA). The facility is supported by the U.S. National Science Foundation through the MRI program (grant nos. CNS-0821258 and CNS-1228778) and the SCREMS program (grant no. DMS-0821311), with additional substantial support from the University of Maryland, Baltimore County (UMBC). See www.umbc.edu/hpcf for more information on HPCF and the projects using its resources.

We also thank the NASA Center for Climate Simulation (NCCS) for the use of their computational resources (Discover). The authors also thank Amy Houghton at USRA for her proofreading of the revised manuscript We also gratefully acknowledge the two anonymous reviewers, whose remarks greatly improved the quality of this article.

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

**Figures**

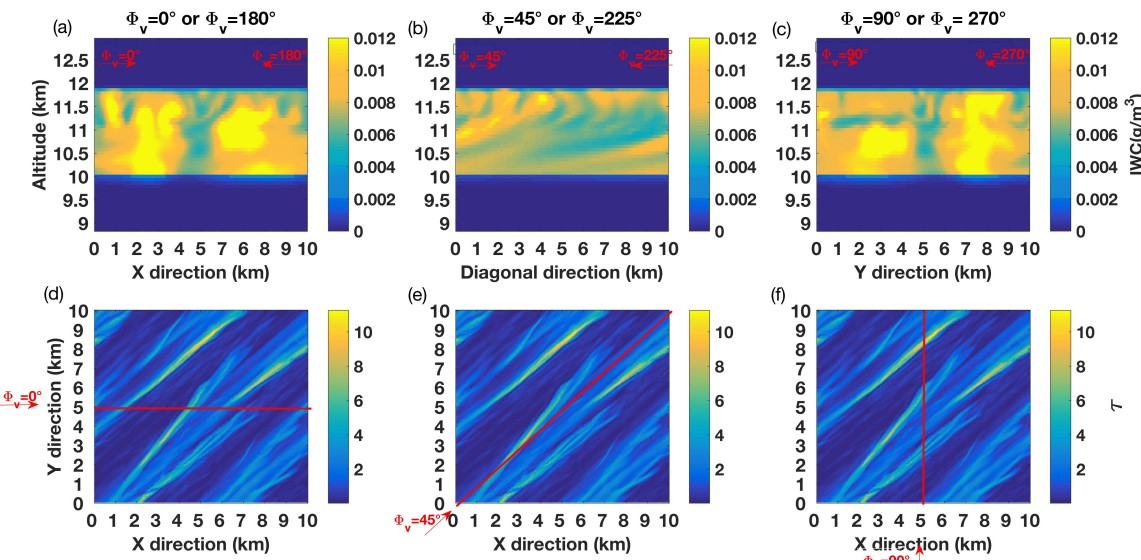

**Figure 1.** Top panels ((a), (b) and (c)) vertical distribution of the ice water content (IWC $(g/m^3)$) following the red lines through the 50 m spatial resolution optical thickness field shown in the bottoms panels ((d), (e) and (f)) along the red line as a function of the azimuth viewing angle $\Phi_v$.)

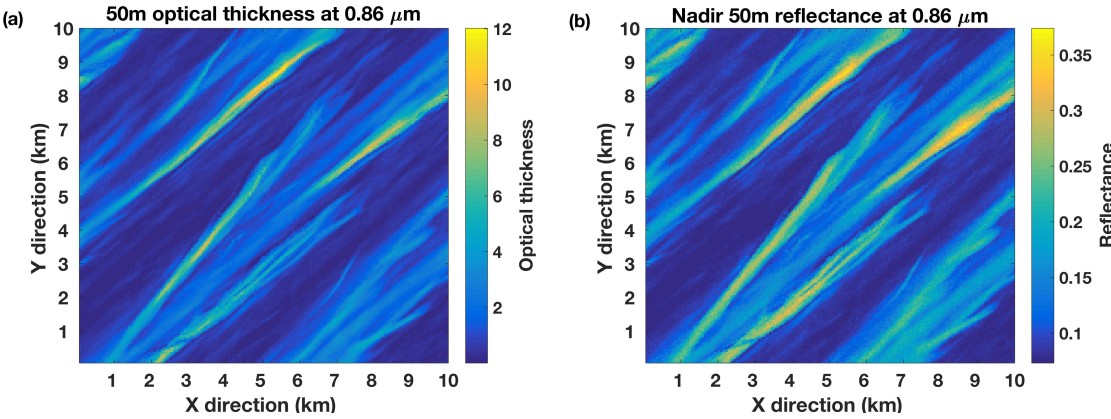

**Figure 2.** (a) Optical thickness field at 0.86 $\mu m$ and (b) solar reflectance field at 0.86 $\mu m$ at a spatial resolution of 50 m, with nadir view and overhead Sun.

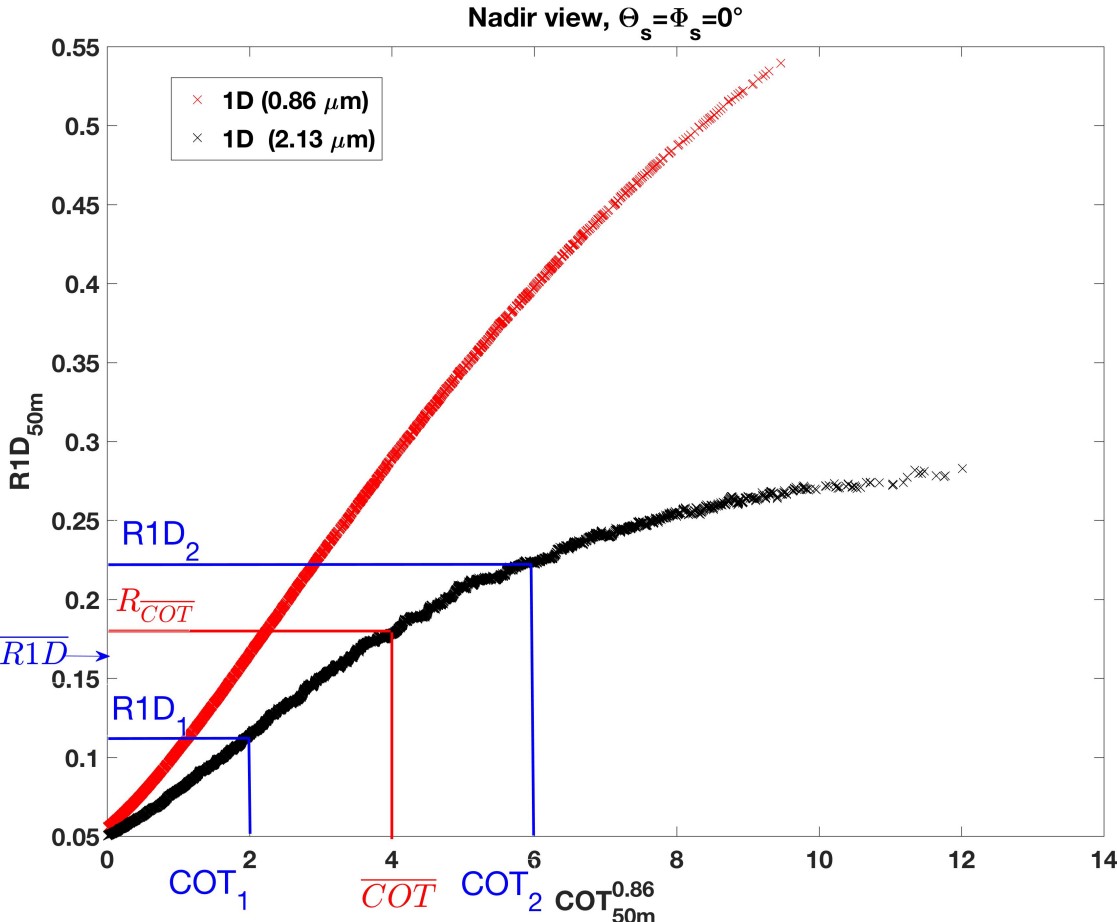

**Figure 3.** 1D reflectances ($R1D_{50m}$) as a function of the 50 m optical thickness at $0.86\ \mu m$ ($COT_{50m}^{0.86\mu m}$) for channels centered at $0.86\ \mu m$ (red) and $2.13\ \mu m$ (black) for nadir view and overhead Sun. $\overline{R1D_1}$ and $\overline{R1D_2}$ represent the reflectances corresponding to the optical thicknesses $COT_1$ and $COT_2$, respectively, and $\overline{COT}$ is their averaged value. Because of the non-linearity between $R1D$ and $COT$, the average reflectance $\overline{R1D}$ is smaller than the reflectance of the average optical thickness $R_{\overline{COT}}$.

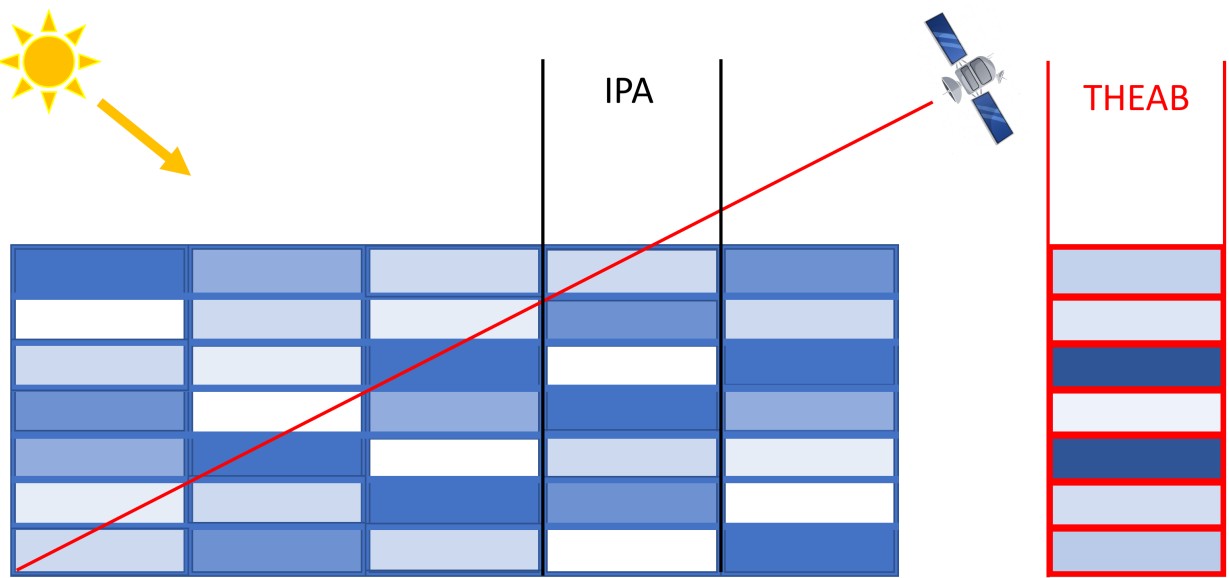

**Figure 4.** Schematic illustration of the Tilted Homogeneous Extinction Approximation used for calculating the Tilted Homogeneous Extinction Approximation Bias (THEAB). The line of sight crosses various cloudy columns with variable extinctions while the Independent Pixel Approximation (IPA) considers only the cloudy column directly under the observed pixel at the top of the cloud. Note that each cell in the THEAB looks darker than in the IPA because we account for the longer path through the cell while keeping the cell size constant which implies to increase the extinction.

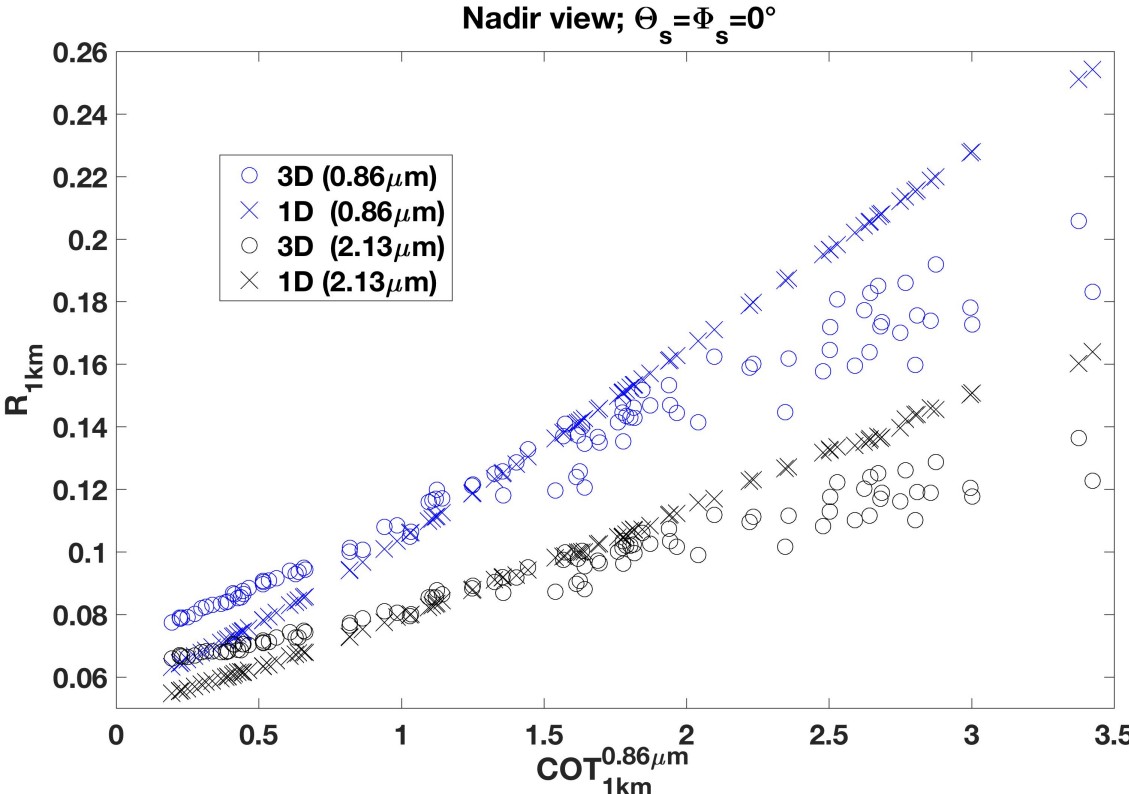

**Figure 5.** Solar reflectances at $0.86$ $\mu m$ (blue) and $2.13$ $\mu m$ (black) at 1 km spatial resolution computed in 3D (circles) and in 1D (crosses). We choice to show these reflectances at 1 km to not overload the plot with too many points at finer spatial resolutions but the effect is the same.

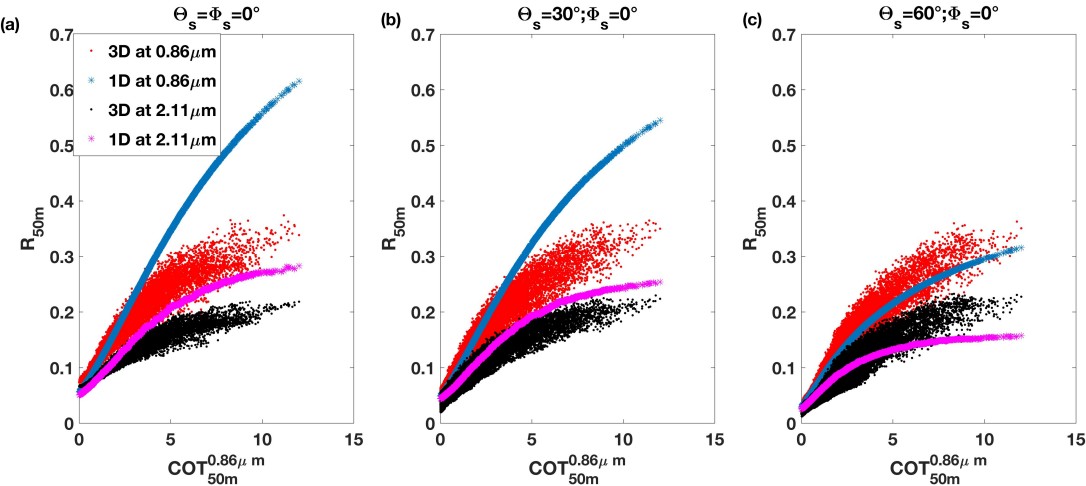

**Figure 6.** Reflectances ($R_{50m}$) for nadir view, as a function of 50 m optical thickness for channels centered at $0.86\ \mu m$ (in red and blue colors for 3D and 1D computations, respectively) and $2.13\ \mu m$ (in black and magenta colors for 3D and 1D computations, respectively) plotted separately for the three solar zenith angles $\Theta_s$.

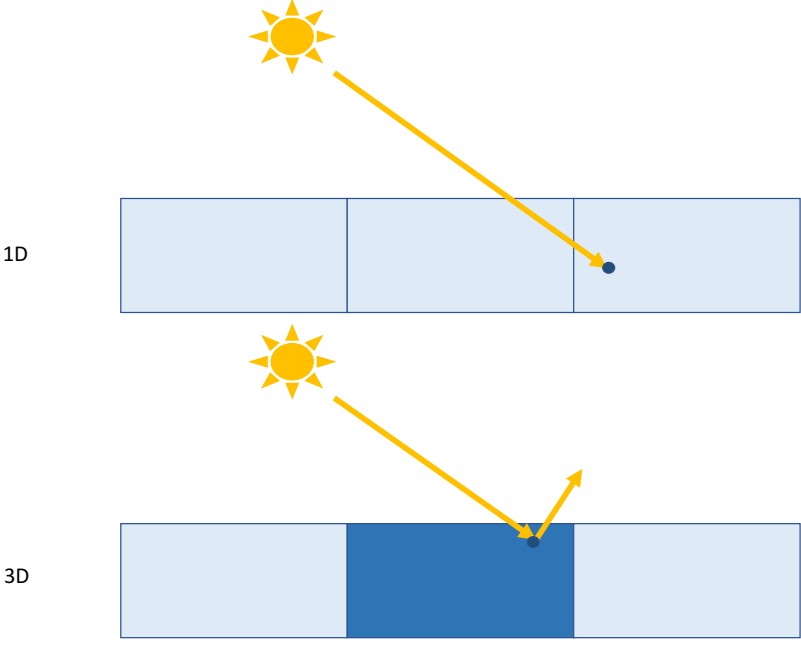

**Figure 7.** Illustration of the shadowing effect. In 1D (top panel), the right column can be illuminated by the photon coming from the Sun, while in 3D (bottom panel), an optically thick neighbor region scatters the photon first , increasing the reflectance of the thick region, but reducing the reflectance of the thin region.

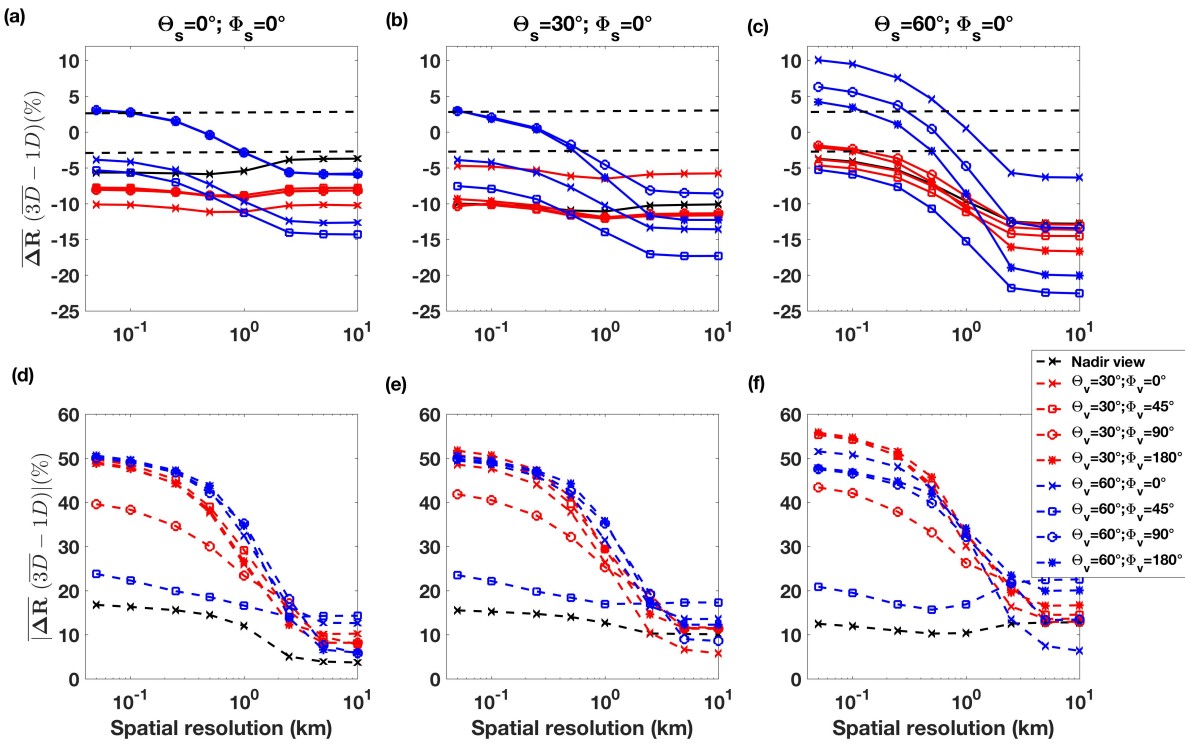

**Figure 8.** Arithmetic ($\overline{\Delta R}$) and absolute ($\overline{|\Delta R|}$) differences between 3D and 1D reflectances at $0.86\ \mu m$ relative to the 3D reflectances in percentage, estimated with equations 1 (panel (a), (b) and (c)) and 2 (panel (d), (e) and (f)), respectively, as a function of the spatial resolution. Each line is for a different pair of viewing zenith and azimuth angles $\Theta_v$ and $\Phi_v$, and each panel is for a different solar zenith angles $\Theta_s$. The horizontal dashed lines represent the MODIS reflectance measurement uncertainty of $3\%$ (Xiong et al., (2005, 2017)).

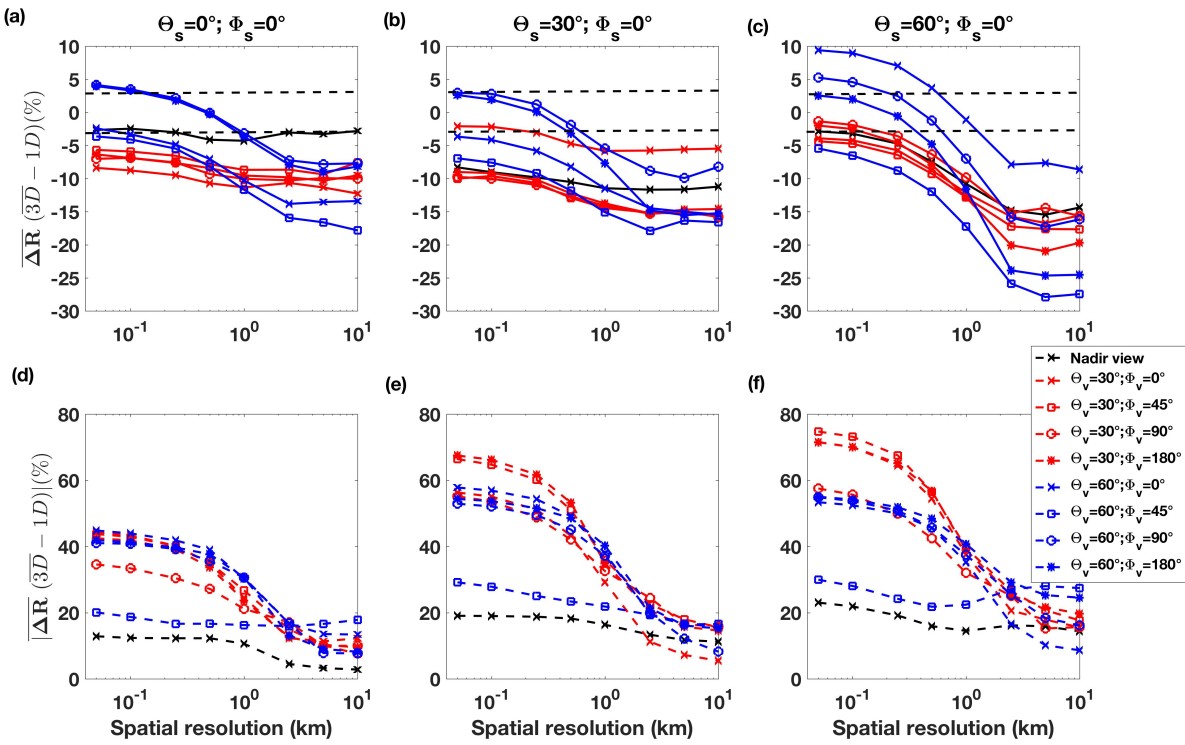

**Figure 9.** Arithmetic ($\overline{\Delta R}$) and absolute ($\overline{|\Delta R|}$) differences between 3D and 1D reflectances at 2.13 $\mu m$, relative to the 3D reflectances in percentage, estimated with equations 1 (panel (a), (b) and (c)) and 2 (panel (d), (e) and (f)), respectively, as a function of the spatial resolution. Each line is for a different pair of viewing zenith and azimuth angles $\Theta_v$ and $\Phi_v$, and each panel is for a different solar zenith angles $\Theta_s$. The horizontal dashed lines represent the MODIS reflectance measurement uncertainty of $3\%$ (Xiong et al., (2005, 2017)).

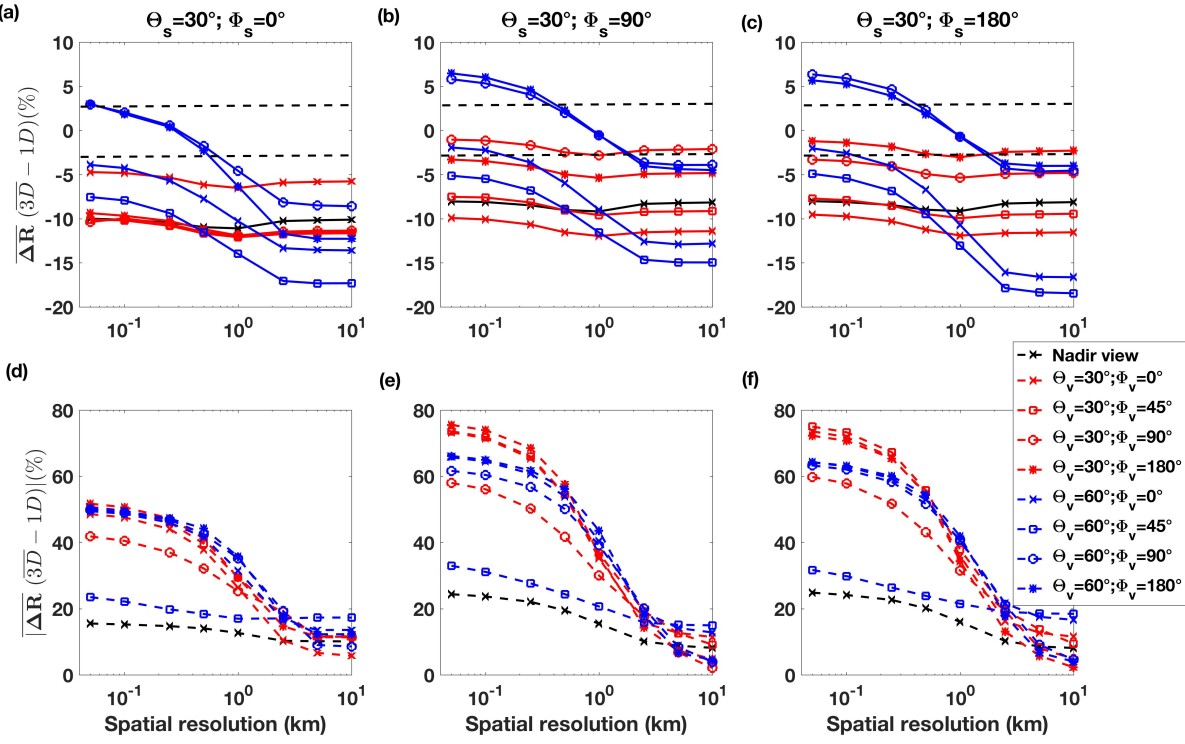

**Figure 10.** Mean arithmetic ($\overline{\Delta R}$) and absolute ($\overline{|\Delta R|}$) differences between 3D and 1D reflectances at $0.86\ \mu m$, relative to the 3D reflectances in percentage, estimated with equations 1 (panel (a), (b) and (c)) and 2 (panel (d), (e) and (f)), respectively. Each line is for a different pair of viewing zenith and azimuth angles $\Theta_v$ and $\Phi_v$, and each panel is for a different solar azimuth angles $\Phi_s$ for solar zenith angle $\Theta_s = 30°$. The horizontal dashed lines represent the MODIS reflectance measurement uncertainty of $3\%$ (Xiong et al., (2005, 2017)).

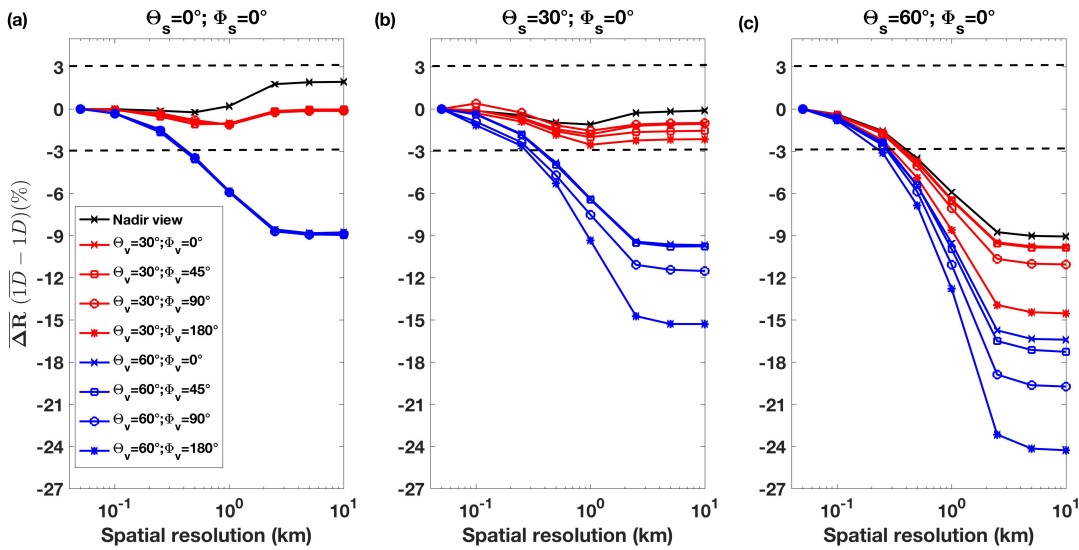

**Figure 11.** Plane-Parallel and Homogeneous Bias (PPHB) representing by the arithmetic $(\overline{\Delta R}(\overline{1D}-1D))$ differences in percentage between 1D reflectances at $0.86 \ \mu m$ relative to the 3D reflectances in percentage, estimated with Eq. 3 as a function of the spatial resolution for various viewing zenith and azimuth angles $\Theta_v$ and $\Phi_v$ , respectively (black, red and blue colors), and solar zenith angles $\Theta_s$ (from the left to the right panels). The horizontal dashed lines represent the MODIS reflectance measurement uncertainty of $3\%$ (Xiong et al., (2005, 2017)).

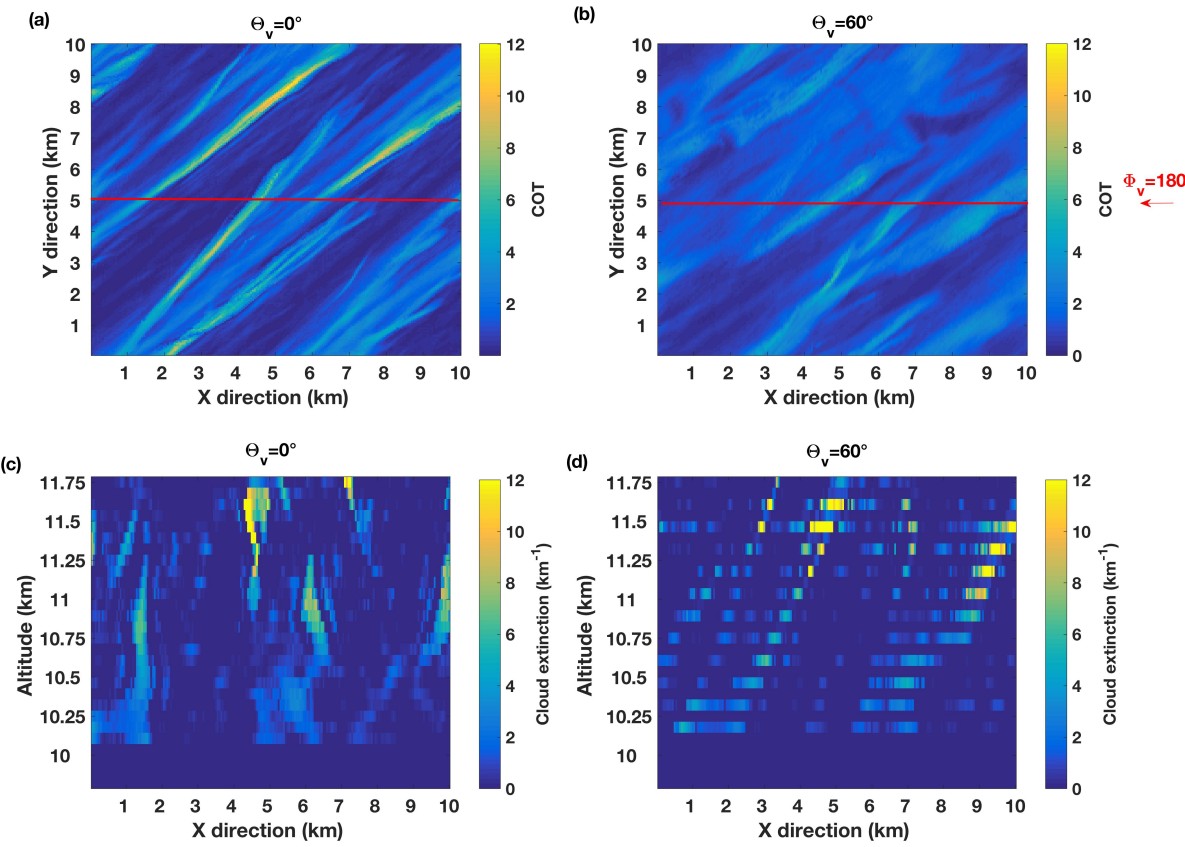

**Figure 12.** Optical thickness (COT) field view from nadir (a) and from a view zenith angle $\Theta_v = 60°$ and vertical profile of the cloud extinction coefficient view from nadir (c) and from $\Theta_v = 60°$ (d) along the red line of the COT field in (a) and (b), respectively for a viewing azimuth angle $\Phi_v = 180°$.

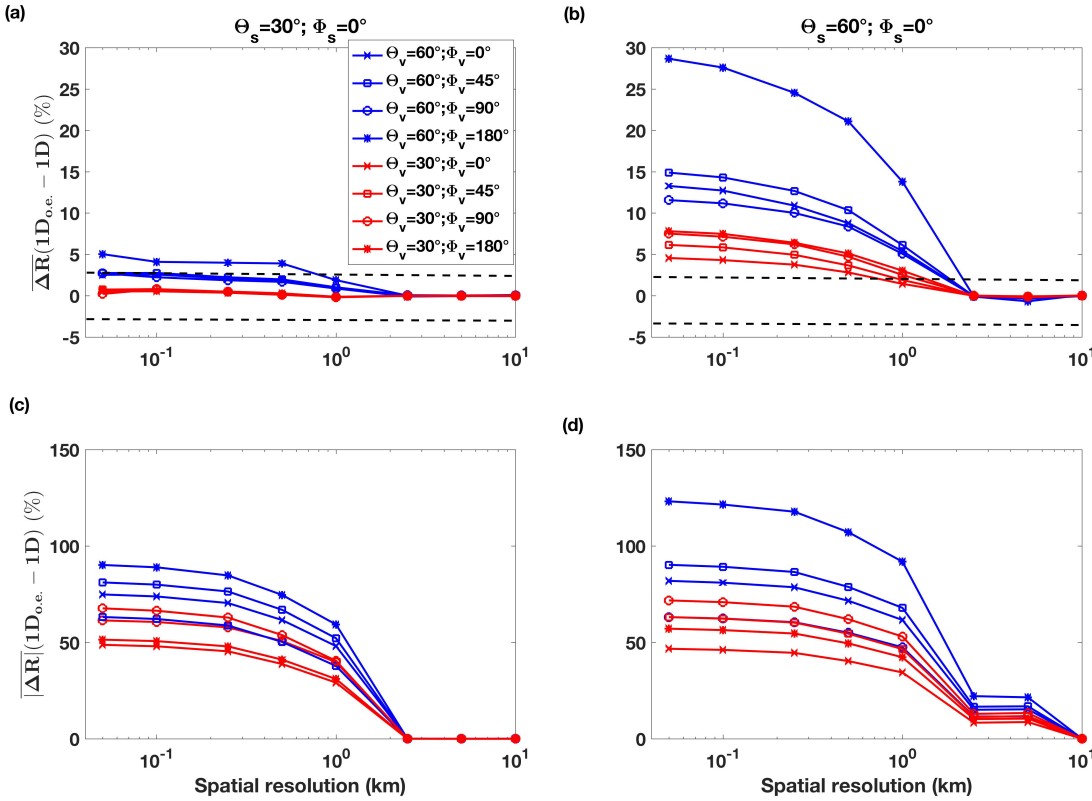

**Figure 13.** Tilted and homogeneous extinction approximation bias (THEAB) relative to the 3D reflectances in percentage, estimated by Eq. 4 (panels (a) and (b)) and 5 (panels (c) and (d)) for solar zenith angles of $\Theta_s = 30°$ and $\Theta_s = 30°$ and $180°$ solar azimuth angle, for several viewing zenith and azimuth angles $\Theta_v$ and $\Phi_v$. The horizontal dashed lines represent the MODIS reflectance measurement uncertainty of $3\%$ (Xiong et al., (2005, 2017)).

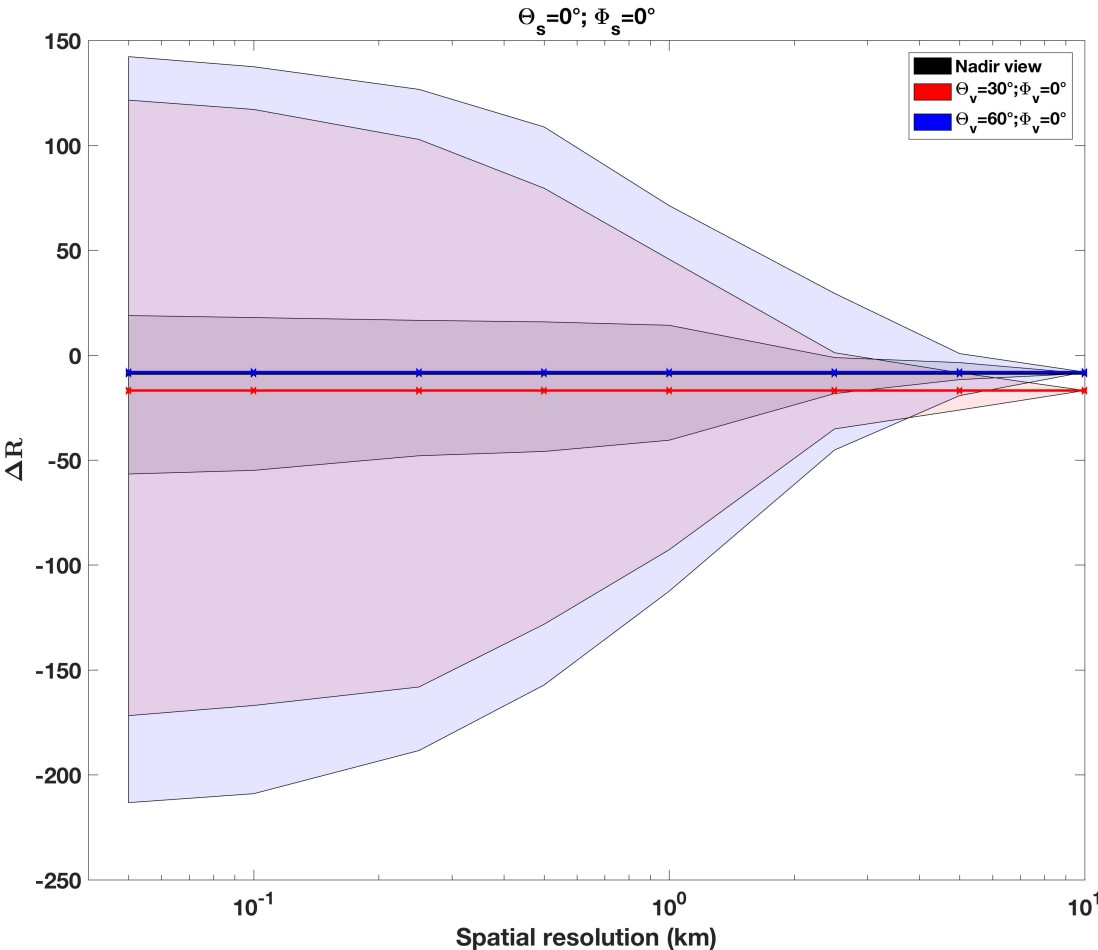

**Figure 14.** 3D effects as a function of the spatial resolution relative to the 3D reflectances in percentage. The shade areas correspond to the range of relative reflectance differences estimated for the 10th and 90th percentiles and the solid lines correspond to the average differences for different pair of viewing zenith and azimuth angles $\Theta_v$ and $\Phi_v$.

**Tables**

**Table 1.** Optical properties (extinction coefficient $\sigma_e$, single scattering albedo $\varpi_0$ and asymmetry parameter of the phase function $g$) of the ice crystal distribution used in this study, which assumes an effective radius of 10 $\mu m$ and an aggregate column shape provided by the Yang et al. (2013) model

| | $\sigma_e$ | $\varpi_0$ | g |
|---|---|---|---|
| MODIS channel 2 (0.86 $\mu m$) | 2.086 | 0.9999855 | 0.7526803 |
| MODIS channel 7 (2.13$\mu m$) | 2.100 | 0.9621367 | 0.7898260 |

**Tables**

**Table 2.** 3D effects, in percentage relative to the 3D reflectance, averaged over the 10 km field for various viewing zenith and azimuth angles (resp. $\Theta_v$ and $\Phi_v$) and solar zenith and azimuth angles (resp. $\Theta_s$ and $\Phi_s$).

| | Nadir | $\Theta_v = 30°$ $\Phi_v = 0°$ | $\Theta_v = 30°$ $\Phi_v = 45°$ | $\Theta_v = 30°$ $\Phi_v = 90°$ | $\Theta_v = 30°$ $\Phi_v = 180°$ | $\Theta_v = 60°$ $\Phi_v = 0°$ | $\Theta_v = 60°$ $\Phi_v = 45°$ | $\Theta_v = 60°$ $\Phi_v = 90°$ | $\Theta_v = 60°$ $\Phi_v = 180°$ |
|---|---|---|---|---|---|---|---|---|---|
| Zenith | -8.7 | -16.9 | -12.9 | -13.4 | -13.4 | -8.1 | -11.1 | 6.4 | 6.2 |
| $\Theta_s = 30°$ | | | | | | | | | |
| $\Phi_s = 0°$ | -14.9 | -7.2 | -15.7 | -16.0 | -14.6 | -7.6 | -14.6 | 5.8 | 7.0 |
| $\Phi_s = 90°$ | -11.6 | -15.3 | -11.7 | -1.6 | -5.1 | -3.8 | -10.0 | 11.3 | 12.9 |
| $\Phi_s = 180°$ | -11.5 | -14.8 | -11.9 | -5.1 | -1.9 | -4.7 | -10.9 | 12.7 | 11.1 |
| $\Theta_s = 60°$ | | | | | | | | | |
| $\Phi_s = 0°$ | -3.9 | -4.3 | -5.2 | -2.2 | -2.9 | 15.2 | -8.0 | 10.5 | 9.8 |