# Peer review of "Scale dependence of cirrus heterogeneity effects. Part II: MODIS NIR and SWIR channels"

_Atmospheric Chemistry and Physics, 2018_

## Referee Comment (RC1) · Anonymous Referee #1 · 10 Apr 2018

**1  General Comments**

In this paper, the authors performed a set of 3D and 1D radiative transfer simulations with varying solar and measurement viewing angles. The goal of the study is to characterize and disentangle the various effects, problems and shortcoming that stem from 1D compared to 3D radiative transfer. The simulations seem appropriate for this endeavor, the results seem reasonable and I liked the review of possible 3D effects.

However, I think the manuscript falls a bit short on the discussion and future implications of the results. The manuscript could include the following two key points before I would want to recommend to publish it.

1. Could you please discuss the magnitude of errors that you find in reflectances compared to current satellite observations? I guess that todays reflectance measurements are on the order of 5 to 10%? How does that relate to or impact your findings?

2. I think the manuscript would greatly benefit from a short discussion or sensitivity test regarding the impact of your findings on actual retrievals. Could you for example provide estimates for errors in retrieved effective radius. I know you promise a future study grounded on optial estimates with spectral information from all ranges but could you not provide or discuss first estimates for a retrieval like Nakjima-King?

**1.0.1 Specific Comments**

- *p*.4 l.20-26 Please state if the domain has cyclic boundary conditions. I guess this is important because the interaction radius may be quite far?

- *p*.4 l.28-33 Please give a more detailed description of the simulation so that one could reproduce your setup. What is the surface albedo? Was aerosol used? Water vapor background profile?

- *p*.5 l.24 IPA does not necessarily mean that the scene is vertically homogeneous. 1D RT is very well capable of simulating vertically inhomogeneous atmospheres numerically. This phrasing of yours kept me wondering till the end of the manuscript if you actually averaged the ice water content vertically or not. Please make that more clear.

- *p*.9 l.4 I think a schematic would be very helpful for the tilted part. I am wondering which slanted path you used, i.e. did you take the optical properties along the sun angle or from viewing geometry. Out of the two, which one would you think

is better? Also, did you use an interlaced grid like for example Wissmeier2013 https://doi.org/10.1175/JAMC-D-12-0227.1?

- *p.*10 l.21-26 Would it not make sense to have a look at 45°and 45°+90°? Is there a particular reason you did not examine that?

- *p.*12 l.19-21 Isn't that particularly interesting for retrievals that use both channels? Wouldn't Nakajima King for example suffer from this even if there would be a linear relationship between the errors in those two channels?

**1.0.2  Minor remarks**

- *p.*1 l.15 "by" should be with?

- *p.*2 l.12 "thicknesses" should be singular

- *p.*5 l.01 shown "in" table

- *p.*5 l.16 I assume you meant 100e9 for all simulations? This is impressive if that is a single core performance which would be 2e6 photons per sec. Or was that parallelized on multiple cores/nodes? If so please state the number of core hours.

- *p.*5 l.27 Conversely?

- *p.*7 l.12 "more" should be "higher"?

- *p.*7 l.16 + fig.2 Please change the color of the 1D markers and put them on top, I could not see your claims.

- *p.*7 l.20 effects should be singular?

- *p.*8 l.24 green? . . . and I thought I am not colorblind. . .

- *p*.9 l.02 temperatures should be singular

- *p*.10 l.7 remove "issue"?

- *p*.11 l.6 remove "view"?

- *p*.11 l.34 remove the "a" in "for a various view"

- *p*.12 l.08 process should be plural

- *p*.12 l.09 Simulation should be plural?

- *p*.12 l.12 take should be taken

- *p*.12 l.13 maybe change "here are ranged to" "here range from"

- *p*.12 l.21 "shown" should be "show"?

- *p*.13 l.06 "this" should be "these"?

- *p*.13 l.20 insert "from" after "ranging"

- *f*ig.1 Error in caption, reference to (f) is (e)

- *f*ig.2 if you mention 50m here I am wondering was it something else in fig1?

- *f*ig.3 I would like it very much if you could provide short conclusions here already in the caption

- *f*ig.4 As mentioned earlier, please update the colors so that a reader can distinguish the markers

- *f*ig.5 please add the idea of the panels and colors to the caption. I.e. left to right are zenith angles, colors are view zenith angles

- *f*ig.6 put brackets around equation references? Change null to $\Phi_s = 0$

- *t*able 1 MOD06 optical properties please write out the names and symbols

- *t*able 1 why use diameter here when you use radius everywhere else?

- *t*able 1 $\lambda$ for channel 2 differs. . . is $.83$ micron correct?

---

## Referee Comment (RC2) · Anonymous Referee #2 · 19 Apr 2018

In principle, this manuscript will make a contribution because the 3D effects of cirrus have not been studied very extensively. It essentially seeks to translate findings of earlier studies by Zinner, Davis and others, which were done for low clouds, to thin, high clouds. The issue with the manuscript in its current state is that language shortcomings make it very hard to follow, particularly in section 4. The manuscript overall reads like a draft that has not been vetted by the co-authors. Some of the figures speak for themselves, but the text tends to confuse in many places, rather than guiding the readers' eyes. The interpretation of 3D effects is also questionable (see comments #1, #10, #11). Given the multitude of typos, grammatical errors, and non-idiomatic or semantically wrong use of the language (for which examples are provided below), I recommend to reconsider the manuscript after major revisions, or reject it to give

the authors more time to edit. While it was not possible to give this a full review for the aforementioned reasons, the factual content does seem promising - with a few reservations listed below. The only major ones are #1, #2, and #10.

1) p7, l11: "photons in thin columns have less chance to be absorbed" . . . "photons in neighboring columns with stronger scattering have more chance to leave the cloud if they are scattered toward a neighboring column with smaller extinction coefficient".

This seems to advocate for the flawed notion of photons moving along contrasts in extinction coefficients or optical thickness, a common misconception that does not pass muster upon closer examination. Perhaps a few figures showing the spatial distribution of some of the discussed biases would elucidate this issue.

2) p4, l31: Is a CER of 10 micron really representative? It's very small, although not outside the climatology. Even in collection 6, 30 micron is the median value of the global distribution of ice clouds.

3) p3,l34: ". . .because side illumination and shadowing almost cancel out each other, there is an overall agreement between CER retrieved using 2.1 or 3.7 micron."

This is unclear: is that relative to 1.6 micron? Why do "side illumination" and "shadowing" cancel each other? Does that refer to the domain average, and if so, over how large a domain does one need to average?

4) p5,l21: At the beginning of this section, a more general description of 3D effects and their differences in the thermal and solar range would be in order. The "3D paths" that "radiation follow" [sic] are associated with fundamentally different physics, which deserves a thorough discussion. For example, scattering is much less important in the thermal wavelength range. This paper quickly dives into the details without providing a more general overview first. Furthermore, the observed dependencies on scale deserve a thorough justification.

5) [abstract] "This strong wavelength dependency [sic] of cirrus cloud radiative effects"

Does this refer to the contrast between solar and thermal IR bands?

6) p7,l15-17: The figure does not support this explanation. Isn't there a much simpler one? For lower sun elevation, satellites are more likely to pick up side scattering than for high sun elevation, especially for optically thin clouds. While this is in the realm of speculation, the explanation by the authors, finding different effects in different optical thickness ranges, is not supported by the figures. See also comment 1.

7) In many places, the manuscript talks about an "increase" or "decrease" of reflectance without specifying the direction (e.g., p7,l21). It is important to include this information because 3D effects redistribute radiation differently - which can lead to a reflectance enhancement in one direction, and a decrease in another.

8) Eq. 1: On the left hand side, there is a difference between a quantify with index "R" (reflectance) and a quantity with index "tau" (optical thickness). While not explained, it is assumed that the latter really means the reflectance calculated for a certain "tau", but the use of a retrieval parameter on par with a reflectance is a bit confusing, as is the nomenclature of the formulae in general. Simplifications would help tremendously.

9) "PPHB increases as the spatial resolution increases": This is misleading throughout the manuscript. What is meant here is "aggregation pixel size", not spatial resolution. Higher spatial resolution actually means a smaller size of the individual pixels.

10) p8,l25: "Note that for sun at zenith . . . [sic]". When speaking about the PPHB in particular, it is hard to see why the sun angle would have an impact. Isn't the argument here that the optical thickness is small, which means that the retrievals are done in the linear (non-asymptotic) range of the LUT? PPHB is ultimately due to the morphology of the LUT, so it is hard to picture a role for SZA.

11) "The THEAB is therefore a consequence of the PPHB for oblique view". The statement before does not support this assertion. The PPHB is fundamentally different from IPA/THEA; the latter two, on the other hand, are related.

12) p9,l16: This seems to be a somewhat unfortunate description of a version of TIPA. Would it be easier to just refer to one of the TIPA papers - for example, Várnai 99?

13) p10,l8: Why does "non-aggregated" have coarser resolution? Isn't it just the opposite?

Summarizing, the factual problems seem to lie in a rather superficial interpretation of the findings, and they could benefit from discussion with co-author Tamas Varnai and other experts in the field. The problems are compounded by many language errors, and I advise to run a spell and grammar check, and further to go through punctuation and semantic/idiomatic use of words. Such issues are not within the purview of manuscript reviewers. The time spent on this review is somewhat out of proportion with the current overall level of maturity of the manuscript.

Examples in no particular order:

Figure 8 caption: In 1D (top panel) [missing comma] the right column can be highlighted [should be "illuminated" - semantic error] by the photon coming from the Sun [missing comma] while in 3D, a [an optically] thick neighbor region intercept [intercepts?] first the photon [first intercepts ?] and scatted [scattered] it back to space. Aside from the errors, this statement is also hard to understand. Also, what is the difference between "intercept" and "scatter"? Physically accurate would be "scatter" or "attenuate".

"the variety of voxel extinctions from a line of sight to another can be quite similar". (In this case, it's unclear what this means - perhaps that the extinction along the line of sight varies little from one tilted column to the next?)

"This is because of the THEAB which is a positive bias, stronger at high resolutions and large view angles." Not a sentence.

"are more highlighted from the side" - this should be "illuminated" throughout the manuscript, unless the intention was to say "highlight", but that doesn't seem to make sense.

"depriving neighbor cloudy columns from [of] incoming photons" – aside from the wrong preposition, using "deprive" seems inappropriate for an inanimate object.

"an important factor that constrains the impact of these assumptions" - "determines" instead of "constrain"?

"To compare reflectances issue from a 3D radiative transfer. . ." use of "issue" is unclear [as noun or verb]

"conversevely" [sic] - several such typos that a spell checker would pick up

"have an almost nil effect" - wrong semantic context for "nil"

"which becomes almost null" - zero? idiomatic/semantic error

"since less different cloudy columns are crossed" - it should be "fewer" instead of "less"

"the PPHB increases as the spatial resolution increases" (the intended phrasing was probably: "the PPHB increases as the spatial resolution decreases (pixel size/aggregation level increases)".

"the absolute 3D effects are slightly smaller and follow the same decreasing with coarsening spatial resolution" - "follow the same decreasing with coarsening" does not seem to work. Perhaps "also decreases with coarser resolution"?

"fallstreaks or not" - "whether fallstreaks are included or excluded"?

W.m-2: What is the meaning of the dot - found throughout the manuscript?

"can be extrapolated to other cirrus clouds" - "generalized" instead of "extrapolated"?

"spatial resolutions considered here are ranged from . . ." - spatial resolutions considered here range from . . .

"most of the figures shown [showed] the"

"THEAB and PPHB is [are] complicated"

"view zenith angle" - "viewing" instead?

"because of no THEAB" - because THEAB is turned off [or some qualifier instead of a "no"]

p2,l22-23: What is the difference between "information content" and "retrieval methods" in this case? They are two different categories.

p4,l2: "LES domain" - was LES introduced before?

---

## Author Comment (AC1) · 12 Jul 2018

**Reply to Anonymous Referee #1**

*1 General Comments*
*In this paper, the authors performed a set of 3D and 1D radiative transfer simulations with varying solar and measurement viewing angles. The goal of the study is to characterize and disentangle the various effects, problems and shortcoming that stem from 1D compared to 3D radiative transfer. The simulations seem appropriate for this endeavor, the results seem reasonable and I liked the review of possible 3D effects. However, I think the manuscript falls a bit short on the discussion and future implications of the results. The manuscript could include the following two key points before I would want to recommend to publish it.*

We would like to thanks the anonymous referee #1 for his careful look at our paper. In the text below we answer point by point to anonymous referee #1 comments and questions. We have also included the necessary changes in the manuscript. Also, we changed the way to plot the figures, now the various effects showed in the manuscript are relative (in %) to the 3D reflectances (the truth) at the given spatial resolution. We have also over-imposed the MODIS reflectance accuracy ~3% to each of these plots in order to show when the impact of 3D and/or heterogeneity on the reflectances are significant or not.

In addition, to be clearer for explaining the cloud heterogeneity, we change the structure of the paper by first presenting the total differences between 3D and 1D reflectances and then the PPH bias, the THEAB and the 3D effects.
We also add Fig. 4 and Fig. 5 to illustrate the THEAB and 3D effects, respectively
We added a new section 4.4 on the 3D effects with a new figure (Fig. 14) and a table (Table 2). The conclusions has been deeply re-written.

Also our manuscript has been proofreaded by a native English speaker.

*1. Could you please discuss the magnitude of errors that you find in reflectances compared to current satellite observations? I guess that todays reflectance measurements are on the order of 5 to 10%? How does that relate to or impact your findings?*
This is a very pertinent remark, indeed we agreed that it would be valuable for the reader to compare the impact of 3D and heterogeneity effects to the reflectance measurements accuracy.
Currently for MODIS, such accuracy is estimated at 3% (Xiong et al., 2005; 2017). All the figures discussion about the amplitude of 3D and heterogeneity effects are now in percentage in order to be compared to the retrieval accuracy and some discussions are added comparing the measurements accuracy and the magnitude of cloud heterogeneity impacts.

*2. I think the manuscript would greatly benefit from a short discussion or sensitivity test regarding the impact of your findings on actual retrievals. Could you for example provide estimates for errors in retrieved effective radius. I know you promise a future study grounded*

*on optial estimates with spectral information from all ranges but could you not provide or
discuss first estimates for a retrieval
like Nakjima-King?*
We agreed with referee #2 that this is an important point to show the impact on the optical
property retrieval, however these results have already published in (Fauchez, T., Platnick, S.,
Sourdeval, O., Meyer, K., Cornet, C., Zhang, Z., and Szczap, F. Cirrus heterogeneity effects on cloud
optical properties retrieved with an optimal estimation method from MODIS VIS to TIR channels.
AIP Conference Proceedings, 1810(1): 10 040002, 2017b.) at a 1km spatial resolution and
submitted in (Fauchez, T., Platnick, S., Sourdeval, O., Wang, C., Meyer, K., Cornet, C., and Szczap.
F. *Cirrus horizontal heterogeneity and 3D radiative effects on cloud optical property retrievals
from MODIS near to thermal infrared channels as a function of spatial resolution)* in the upcoming
JGR special issue: "3D Cloud Modeling as a Tool for 3D Radiative Transfer".
In this study we have shown that when only considering cloud horizontal heterogeneity effects,
the largest retrieval errors are associated with TIR retrievals due to the PPH bias. However, when
both cloud 3D and heterogeneity effects are considered, the solar reflectance-based retrievals
have the largest error for spatial resolutions less than 500–1000 m, while the TIR-based retrievals
have the largest error above this resolution due to PPH bias.
COT and CER retrieval errors using SWIR/VNIR reflectance measurements are of the order of 10
% for a Sun at zenith but they can be up to 20 % [at 10 km spatial resolution] and 100%  [ at 50 m
spatial resolution] for COT and CER retrievals, respectively, for an oblique Sun.

**1.0.1 Specific Comments**
*• p.4 l.20-26 Please state if the domain has cyclic boundary conditions. I guess this is
important because the interaction radius may be quite far?*
Yes the domain has cyclic boundary conditions.
We add in the revised manuscript, line 24 page 4 the following sentence: "… *assuming cyclic
boundary conditions are imposed at the edges of the domain*"

*• p.4 l.28-33 Please give a more detailed description of the simulation so that one could
reproduce your setup. What is the surface albedo? Was aerosol used?Water vapor
background profile?*
The detailed description of the simulations are already presented in the Part I of the paper.
However, we agreed that it is valuable to be presented in this second part too. Ocean surface
albedo values for NIR/SWIR channels are set at 0.05, there is no aerosol and the mid-latitude
summer atmospheric profile is presented in Fig 2 of Fauchez et al., 2017a.
We add line 9, page 5: "*No aerosol is added and the surface is Lambertian with a constant
albedo of 0.05. No aerosol and atmospheric absorption are considered.*"

*• p.5 l.24 IPA does not necessarily mean that the scene is vertically homogeneous.
1D RT is very well capable of simulating vertically inhomogeneous atmospheres numerically.
This phrasing of yours kept me wondering till the end of the manuscript if you actually
averaged the ice water content vertically or not. Please make that more clear.*
You are right, the IPA implies the horizontal inhomogeneity and pixel independence not the
vertical inhomogeneity. However, for the retrieval of optical properties from space radiometers

only one crystal size/shape is retrieved regardless the vertical extension and inhomogeneity of the cloud column. In our study the cloud extinction is vertically inhomogeneous, however only one crystal size and shape is assumed in the whole cloud domain for simplification of the problem and assess the extinction coefficient variability effects only. The impact of the vertical variability of ice crystal size/shape is very interesting to study but out of the scope of this paper.

We have remove the "*vertically inhomogeneous*" of L24, P5.

We have added this sentence in the section presenting the microphysical model: "Note that while the microphysical properties are homogeneous, the extinction coefficient varies horizontally and vertically."

**• p.9 l.4 I think a schematic would be very helpful for the tilted part. I am wondering which slanted path you used, i.e. did you take the optical properties along the sun angle or from viewing geometry. Out of the two, which one would you think is better? Also, did you use an interlaced grid like for example Wissmeier2013 https://doi.org/10.1175/JAMC-D-12-0227.1?**

We take the optical properties along the viewing geometry, this is different from the Tilted Independent Pixel Approximation (TIPA, Varnai et al. 1999), which have computed the optical properties along the sun angle.

Thank you for providing us this reference that we did not have. The TICA approach (i.e. paNTICA, parameterized nonlocal tilted independent pixel approximation) described in this paper (Wissmeier et al., 2013) is similar to the TIPA of Varnai et al., (1999). In our study we only account for the tilted view of the line of sight crossing various extinction through different cloudy columns while the ice crystal size and shape are constant. We explain our method in page 9, line 13 to 16:" For each pixel we have re-created a 1D cloud for which the vertical extinction is the averaged oblique extinction of the 3D heterogeneous field. In other words, we have run 1D RT using the oblique columns crossed as adjacent vertical cloud layers (i.e., tilted the oblique columns crossed to a vertical column)."

In order to highlight the relationship between THEAB and TIPA, we included the following sentences into the manuscript:

*In essence, THEAB can be considered a variant of the Tilted Independent Pixel Approximation (TIPA) used in earlier studies (e.g., Várnai and Davies, 1999; Wapler and Mayer 2008; Frame et al., 2009;* Wissmeier et al., 2013*), but with the tilting based on the view direction instead of the solar direction. A somewhat similar concept to THEAB was used in Evans et al. (2008), where reflectances were related to cloud properties calculated along the slanted line of sight.*

Also, as you mentioned, a figure will be very helpful, we add it in the manuscript and refer it as Fig. 4 in the new version.

[Figure]

The computation of the THEAB was helpful to understand why the sign of ΔR(3D-1D) depends on the spatial resolution (for Sun at zenith and off-zenith). We explain the change of sign due to the THEAB in section 4.3 Tilted and homogeneous extinction approximation bias (THEAB) :

*"A somewhat similar concept to THEAB was used in Evans et al. (2008), where reflectances were related to cloud properties calculated along the slanted line of sight. Each tilted line of sight crosses large, medium and small extinctions through many different columns leading to an average optical paths similar between each tilted columns and therefore the field of view appears more homogeneous than the one with independent cloudy columns (1D assumption) with small optical thickness juxtaposed to large optical thickness. This effect is shown in Fig. 11 where we can see that the optical thickness field at 50 m spatial resolution view from 60° zenith angle is much smoother than the one see from nadir. We can also see that the extinction plumes are stretched out, spreading and smoothing the cloud extinction over the columns. Indeed, for a voxel horizontal and vertical sizes of 50 m and 72 m, respectively, and a $\Theta v = 60°$, the line of sight reaching the top of a given voxel from its center (see Fig. 4) then cross horizontally 72 × tan(60) ~ 125 m, i.e. two adjacent voxels before reaching the underneath cloud layer. "*

Because this effects occurs also for the Sun at zenith it is not due to the TIPA, therefore we had no need to compute new time expensive computation for TIPA which is implicitly included in the 3D radiative effects.

**• p.10 l.21-26 Would it not make sense to have a look at 45°and 45°+90°? Is there a particular reason you did not examine that?**
There is no particular reason to not look at these angles except that for computational time reasons we had to select a small number of viewing geometries. Indeed, at 45° the line of sight is parallel to the fallstreaks, at 45°+90° the line of sight will be perpendicular to the fallstreaks that would not differ too much to the other viewing azimuth of 90° and 180°.

**• p.12 l.19-21 Isn't that particularly interesting for retrievals that use both channels? Wouldn't Nakajima King for example suffer from this even if there would be a linear relationship between the errors in those two channels?**

This is an interesting question, both are solar channels (0.86 and 2.13 μm) and they produce similar 3D effects (shadowing, side illumination and horizontal transport). However, because the absorption is different between these two channels the amplitude of the effects is also different. Obviously, because the amplitude of the 3D heterogeneity effects is different between 0.86 and 2.13 μm this will impact the cloud optical property retrieval through a Nakajima and King method (or other similar method using a combination of NIR/SWIR channels, see for instance Zhang et al., (2012, 2013), Marshak et al., (2006)). Note that those differences are much smaller than between TIR and NIR/SWIR channels (see Fauchez et al. 2017b). Also. Fauchez et al., (2017a) have shown that for TIR MODIS channels (centered at 8.52, 11.01, 12.03 and 13.36μm), the difference of cloud absorption (and scattering) between those channels leads to different 3D and heterogeneity effects, which later impact the cloud optical property retrieval using thermal infrared channels such as the split-window technique (see also Fauchez et al, 2017b and 2018).

Marshak, A., S. Platnick, T. Várnai, G. Wen, and B. Cahalan (2006), Impact of three-dimensional radiative effects on satellite retrievals of cloud droplet sizes, *J. Geophys. Res., 111*, D09207, doi:10.1029/2005JD006686.

Zhang, Z., Ackerman, A. S., Feingold, G., Platnick, S., Pincus, R., and Xue, H. Effects of cloud horizontal inhomogeneity and drizzle on remote sensing of cloud droplet effective radius: Case studies based on large-eddy simulations. *Journal of Geophysical Research: Atmospheres*, 117(D19), 2012. D19208.

Zhang, Z. (2013), On the sensitivity of cloud effective radius retrieval based on spectral method to bi-modal droplet size distribution: A semi-analytical model, J. Quant. Spectros. Radiat. Transfer, 129, 79–88, doi:10.1016/j.jqsrt.2013.05.033.

**1.0.2 Minor remarks**
**• p.1 l.15 "by" should be with?**
Yes, done

**• p.2 l.12 "thicknesses" should be singular**
Yes, done

**• p.5 l.01 shown "in" table**
Yes, done

*• p.5 l.16 I assume you meant 100e9 for all simulations? This is impressive if that is a single core performance which would be 2e6 photons per sec. Or was that parallelized on multiple cores/nodes? If so please state the number of core hours.*

Thank you for pointing this out, it was a mistake, the computational time is about 3.5 days for 100 billions of photons for each simulations (spread on 2048 batches). Below is a detail of one of the simulations.:

Total time =  3 days, 13 hrs, 14 mins,  2 secs

Mean time per trial   =      0.80 millisecs

Mean time per scatter =    145.60 microsecs

Mean time per batch =    149.83secs

 102400.00 Mtrials completed with 2048batches

We have corrected the computational time in the manuscript and explain that those simulations have been parallelized on 2048 core each.

*• p.5 l.27 Conversely?*

Yes, done

*• p.7 l.12 "more" should be "higher"?*

Yes, done

*• p.7 l.16 + fig.2 Please change the color of the 1D markers and put them on top, I could not see your claims.*

We assume that you mean Fig. 4 and  we made the change in the figure.

*• p.7 l.20 effects should be singular?*

Yes, done

*• p.8 l.24 green? . . . and I thought I am not colorblind. . .*

No, you are not! That was a mistake, we have corrected it according to the color change you asked for this figure.

*•    p.9 l.02 temperatures should be singular*

Yes, done

*• p.10 l.7 remove "issue"?*

Yes, done

*• p.11 l.6 remove "view"?*

Yes, done

*• p.11 l.34 remove the "a" in "for a various view"*

Yes, done

*• p.12 l.08 process should be plural*

Yes, done

*• p.12 l.09 Simulation should be plural?*
Yes, done

*• p.12 l.12 take should be taken*
Yes, done

*• p.12 l.13 maybe change "here are ranged to" "here range from"*
Yes, done

*• p.12 l.21 "shown" should be "show"?*
Yes, done

*• p.13 l.06 "this" should be "these"?*
Yes, done

*• p.13 l.20 insert "from" after "ranging"*
Yes, done

*• f ig.1 Error in caption, reference to (f) is (e)*
Yes, done

*• f ig.2 if you mention 50m here I am wondering was it something else in fig1?*
No, both are at 50 m spatial resolution so we now mention it too in Fig.1.
*• f ig.3 I would like it very much if you could provide short conclusions here already in the caption*
We add: "Because of the non-linearity between R1D and $\tau$, the $\tau$ retrieved from averaged $\overline{R1D}$ is smaller than the averaged $\bar{\tau}$ retrieved from the two R1D."
*• f ig.4 As mentioned earlier, please update the colors so that a reader can distinguish the markers*
Done, thank you.

*• f ig.5 please add the idea of the panels and colors to the caption. I.e. left to right are zenith angles, colors are view zenith angles*
Done, thank you
*• fig.6 put brackets around equation references? Change null to _s = 0*
We changed equations 2 to Eq.2 and changed null to 0.
*• table 1 MOD06 optical properties please write out the names and symbols*
Done, thank you
*• table 1 why use diameter here when you use radius everywhere else?*
We changed it to effective radius of 10 micron
*• table 1 _ for channel 2 differs. . . is .83 micron correct?*
No it should be 0.86 micron, thank you.
REFERENCES

Fauchez, T., 5 Platnick, S., Meyer, K., Cornet, C., Szczap, F., and Várnai, T. Scale dependence of cirrus horizontal heterogeneity effects on TOA measurements – Part I: MODIS brightness temperatures in the thermal infrared. Atmospheric Chemistry and Physics, 17(13):8489–8508, 2017a.

Fauchez, T., Platnick, S., Sourdeval, O., Meyer, K., Cornet, C., Zhang, Z., and Szczap, F. Cirrus heterogeneity effects on cloud optical properties retrieved with an optimal estimation method from MODIS VIS to TIR channels. AIP Conference Proceedings, 1810(1):10 040002, 2017b.

Fauchez, T., Platnick, S., Sourdeval, O., Wang, C., Meyer, K., Cornet, C., and Szczap. F. *Cirrus horizontal heterogeneity and 3D radiative effects on cloud optical property retrievals from MODIS near to thermal infrared channels as a function of spatial resolution. JGR 2018, in review*

Marshak, A., S. Platnick, T. Várnai, G. Wen, and B. Cahalan (2006), Impact of three-dimensional radiative effects on satellite retrievals of cloud droplet sizes, *J. Geophys. Res., 111*, D09207, doi:10.1029/2005JD006686.

Wissmeier, P., Buras, M and Mayer, B. paNTICA: A Fast 3D Radiative Transfer Scheme to Calculate Surface Solar Irradiance for NWP and LES Models *J. Appl. Meteor. Climatol.*, 52(8):1698-1715, Apr. 1977.

Xiong X., J. Sun, A. Wu, K. Chiang, J. Esposite, and W.L. Barnes,  "Terra and Aqua MODIS Calibration Algorithms and Uncertainty  Analysis," Proceedings of SPIE – Sensors, Systems, and Next Generation  of Satellites IX, 5978, 59780V, doi:10.1117/12.627637, 2005

Xiong, X.,  A. Angal, W. Barnes, H. Chen, V. Chiang, X. Geng, Y. Li, K. Twedt, Z.  Wang, T. Wilson, and A. Wu, "Updates of MODIS on-orbit calibration  uncertainty assessments", Proc. SPIE - Earth Observing Systems XXII, 10402, 104020M 2017

Zhang, Z., Ackerman, A. S., Feingold, G., Platnick, S., Pincus, R., and Xue, H. Effects of cloud horizontal inhomogeneity and drizzle on remote sensing of cloud droplet effective radius: Case studies based on large-eddy simulations. *Journal of Geophysical Research: Atmospheres*, 117(D19), 2012. D19208.

Zhang, Z. (2013), On the sensitivity of cloud effective radius retrieval based on spectral method to bi-modal droplet size distribution: A semi-analytical model, J. Quant. Spectros. Radiat. Transfer, 129, 79–88, doi:10.1016/j.jqsrt.2013.05.033.

---

## Author Comment (AC2) · 12 Jul 2018

Reply to Anonymous Referee #2

*In principle, this manuscript will make a contribution because the 3D effects of cirrus have not been studied very extensively. It essentially seeks to translate findings of earlier studies by Zinner, Davis and others, which were done for low clouds, to thin, high clouds. The issue with the manuscript in its current state is that language shortcomings make it very hard to follow, particularly in section 4. The manuscript overall reads like a draft that has not been vetted by the co-authors. Some of the figures speak for themselves, but the text tends to confuse in many places, rather than guiding the readers' eyes. The interpretation of 3D effects is also questionable (see comments #1, #10, #11). Given the multitude of typos, grammatical errors, and non-idiomatic or semantically wrong use of the language (for which examples are provided below), I recommend to reconsider the manuscript after major revisions, or reject it to give the authors more time to edit. While it was not possible to give this a full review for the aforementioned reasons, the factual content does seem promising - with a few reservations listed below. The only major ones are #1, #2, and #10.*

We would like to thank referee #2 for these very helpful comments who has widely contributed to improve the substance and the form of the paper. We also apologize for the numerous grammatical and typo errors. We greatly appreciate the time referee #2 spent for catching them. We ran a careful check through the whole manuscript and have corrected them in the updated version of the paper. Also, we changed the way to plot the figures, now the various effects showed in the manuscript are relative (in %) to the 3D reflectances (the truth) at the given spatial resolution. We have also over-imposed the MODIS reflectance accuracy ~3% to each of these plots in order to show when the impact of 3D and/or heterogeneity on the reflectances are significant or not.

In addition, to be clearer for explaining the cloud heterogeneity, we change the structure of the paper by first presenting the total differences between 3D and 1D reflectances and then the PPH bias, the THEAB and the 3D effects.
We also add Fig. 4 and Fig. 5 to illustrate the THEAB and 3D effects, respectively
We added a new section 4.4 on the 3D effects with a new figure (Fig. 14) and a table (Table 2). The conclusion has been deeply re-written.

Also our manuscript has been proofreaded by a native English speaker.

*1) p7, l11: "photons in thin columns have less chance to be absorbed" : : : "photons in neighboring columns with stronger scattering have more chance to leave the cloud if they are scattered toward a neighboring column with smaller extinction coefficient".*
*This seems to advocate for the flawed notion of photons moving along contrasts in extinction coefficients or optical thickness, a common misconception that does not pass muster upon closer examination. Perhaps a few figures showing the spatial distribution of some of the discussed biases would elucidate this issue.*

We agreed that this sentence was confusing. What we mean is that real photons, of course, zigzag in all directions and some of them go from thin to thick areas and reverse; it is only the net flow that tends to go from thick to thin. As we can see in Varnai and Davis (1999) figure 5. We add this sentence to explain that this is the net flux:
"*Therefore, the net flux of photons tends to flow from thick to thin regions.*"
And we have also added the new figure below to the manuscript in the section HRT to illustrate this effect
We can see that 3D reflectances are larger than 1D for small optical thicknesses while the opposite is true for large optical thicknesses.

[Figure]

**2) p4, l31: Is a CER of 10 micron really representative? It's very small, although not outside the climatology. Even in collection 6, 30 micron is the median value of the global distribution of ice clouds.**
We agreed with reviewer #2 that and effective radius of 10 microns is a small size for cirrus clouds.
The motivation behind the selection of the 10 microns effective radius is to be consistent with the Part I of this study which focuses on thermal infrared channels. Yet, the sensitivity of retrievals in the thermal infrared is often limited to CER below 20 μm. Therefore, for consistency reasons, we have also selected the same effective radius (10 μm) for this study and in the corresponding papers on cloud optical property retrievals (Fauchez et al., 2017b, 2018).
We have added this paragraph to the conclusion:
*"Note that the results do not significantly change with a larger CER for 0.86 μm because the optical properties are fairly constant up to CER of 50 μm but at 2.13 μm the absorption increases with CER leading to stronger PPH and weaker 3D effects (because the mean free path is reduced by the absorption)."*

**3) p3,l34: ": : :because side illumination and shadowing almost cancel out each other, there is an overall agreement between CER retrieved using 2.1 or 3.7 micron."**
**This is unclear: is that relative to 1.6 micron? Why do "side illumination" and "shadowing" cancel each other? Does that refer to the domain average, and if so, over how large a domain does one need to average?**
This is indeed unclear and we apologize for that. We rephrase it:

*"3D radiative transfer effects, such as illumination and shadowing, can produce significant differences between CER retrievals based on 2.1 μm or 3.7 μm reflectances (along with 0.86 μm) for water cloud. Indeed, the authors showed that 3D effects have stronger impacts on CER retrievals based on 2.1 μm than 3.7 μm, leading to positive difference between the two from cloud side illumination and a negative difference from cloud shadowed. However, these two opposite*

*effects cancel each other out on the domain average, leading to an overall agreement between the CER retrievals."*

**4) p5,l21: At the beginning of this section, a more general description of 3D effects and their differences in the thermal and solar range would be in order. The "3D paths" that "radiation follow" [sic] are associated with fundamentally different physics, which deserves a thorough discussion. For example, scattering is much less important in the thermal wavelength range. This paper quickly dives into the details without providing a more general overview first. Furthermore, the observed dependencies on scale deserve a thorough justification.**

We agreed with referee #2 that this section deserves a better explain of the 3D effects and their differences between wavelength ranges. We re-wrote the paragraph as follow:

*"Clouds are complex 3D structures where solar and terrestrial radiations propagate in a three-dimensional space . However, in current retrieval algorithms, for simplification and/or computational reasons, the homogeneous independent pixel approximation (IPA, Cahalan et al. (1994)) is commonly applied: each portion of the observed cloudy scene is sampled in pixels, and each pixel is assumed to be horizontally homogeneous as well as radiatively independent of its neighbors (1D radiative transfer assumption). The sub-pixel horizontal heterogeneity leads to the plane-parallel and homogeneous bias (PPHB) because of the non-linearity between optical properties and radiance/reflectance. The 1D assumption leads to several effects describing below in terms of 3D radiative effects. Both effects (IPA and PPHB) are strongly dependent on the sensor spatial resolution. The sub-pixel heterogeneity effects increase for coarser spatial resolutions, while 3D effects linked to net horizontal photon transport between columns increase for finer spatial resolutions. The range of spatial resolutions for which either the IPA biases or the PPHB are dominant depends on the wavelength. Off course for thermal wavelength no illumination and shadowing effects are present and in addition cloud absorption is much larger for thermal infrared than for solar wavelengths leading to larger PPHB but smaller IPA effect (because of less scattering)."*

**5) [abstract] "This strong wavelength dependency [sic] of cirrus cloud radiative effects". Does this refer to the contrast between solar and thermal IR bands?**

Yes it does and we agreed that it is more accurate and clear to rephrase the beginning of this sentence as:"
*"The contrast of 3D radiative effects between solar and thermal infrared channels.."*

**6) p7,l15-17: The figure does not support this explanation. Isn't there a much simpler one? For lower sun elevation, satellites are more likely to pick up side scattering than for high sun elevation, especially for optically thin clouds. While this is in the realm of speculation, the explanation by the authors, finding different effects in different optical thickness ranges, is not supported by the figures. See also comment 1.**

Thank you for pointing out that the figure did not support our reasoning clearly. To address this, we changed the color scheme of Figure 4 to make it easy to distinguish 1D and 3D results. We also added the following sentences to the end of the discussion on HRT in order to clarify that only Panel a is relevant to our argument about HRT:

*We note that HRT, as described above, dominates only for overhead sun (Fig 4a and 4b). For oblique sun (Fig 4b and especially 4c) the trend reverses as 3D reflectances exceed 1D ones for optical thicknesses larger than about 5 and 3D reflectances are lower than 1D ones for smaller optical thicknesses. Increase of 3D reflectances oblique sun is caused by the side illumination discussed below.*

**7) In many places, the manuscript talks about an "increase" or "decrease" of reflectance without specifying the direction (e.g., p7,l21). It is important to include this information because 3D effects redistribute radiation differently - which can lead to a reflectance enhancement in one direction, and a decrease in another.**

Good point! Accordingly, we replaced the sentence

This effect occurs when photons of the incoming sunlight travel obliquely which globally increases the reflectance of the cloud by comparison to what is expected in the 1D theory (Loeb and Davies, 1996) as we can see Fig.4 (c) for which most of the 3D reflectances are larger than 1D reflectances.

by the following text:

This effect occurs when photons of the incoming sunlight travel obliquely and enter a cloud through its side and top. In contrast to the HRT, side illumination tends to increase reflectance of thicker clouds (Loeb and Davies, 1996) as we can see Fig. 4c, where most of the 3D reflectances are larger than 1D reflectances. We note however, that side illumination can reduce reflectances in some forward scattering directions due to the "upward trapping process illustrated in Fig 5a of Várnai and Davies (1999).

***8) Eq. 1: On the left hand side, there is a difference between a quantify with index "R"
(reflectance) and a quantity with index "tau" (optical thickness). While not explained,
it is assumed that the latter really means the reflectance calculated for a certain "tau",
but the use of a retrieval parameter on par with a reflectance is a bit confusing, as is
the nomenclature of the formulae in general. Simplifications would help tremendously.***
We acknowledge that the formulae are difficult to read. To simplify we:
1. Remove the "tau" subscript
2. Remove the 50m subscript because the averaged reflectances are always averaged from 50m.

***9) "PPHB increases as the spatial resolution increases": This is misleading throughout
the manuscript. What is meant here is "aggregation pixel size", not spatial resolution.
Higher spatial resolution actually means a smaller size of the individual pixels.***
Thank you for pointing this out, this is indeed wrong and may lead to misinterpretation of the results. We have corrected this through the manuscript.

***10) p8,l25: "Note that for sun at zenith : : : [sic]". When speaking about the PPHB in
particular, it is hard to see why the sun angle would have an impact. Isn't the argument
here that the optical thickness is small, which means that the retrievals are done in the
linear (non-asymptotic) range of the LUT? PPHB is ultimately due to the morphology
of the LUT, so it is hard to picture a role for SZA.***

We do not agree with Reviewer #2 on this point. Indeed, the PPHB depend on the non-linearity between reflectance and optical thickness (Jensen inequality). The intensity of this non-linearity (and thus of the PPHB) depends on the optical thickness but also on viewing and solar angles. Also, the cirrus field has an average optical thickness of 1.5 with values going from 0.008 up to 12 at 0.86um. Therefore, for the largest value the PPHB can be very large.
"

***11) "The THEAB is therefore a consequence of the PPHB for oblique view". The statement
before does not support this assertion. The PPHB is fundamentally different from
IPA/THEA; the latter two, on the other hand, are related.***
We agreed that this sentence is at least confusing. We removed it.

***12) p9,l16: This seems to be a somewhat unfortunate description of a version of TIPA.
Would it be easier to just refer to one of the TIPA papers - for example, Várnai 99?***

We do not agreed with referee #2. The TIPA refers to the oblique of sun radiation while the THEA (Tilted and Homogeneous Extinction Assumption) refers to the line of sight.

In order to highlight the relationship between THEAB and TIPA, we included the following sentences into the manuscript:

*In essence ,the Tilted and Homogeneous Extinction Approximation (THEA) can be considered a variant of the Tilted Independent Pixel Approximation (TIPA) used in earlier studies (e.g., Várnai and Davies, 1999; Wapler and Mayer 2008; Frame et al., 2009), but with the tilting based on the view direction instead of the solar direction. A somewhat similar concept to THEA was used in Evans et al. (2008), where reflectances were related to cloud properties calculated along the slanted line of sight.*

We also add in the end of section 4.3 :
*Note that we choice to calculated the THEAB instead of the TIPA bias because only the former helps to understand why $\Delta R$ is positive for the small scales and negative for the larges, even when the Sun is at zenith (no TIPA bias). The TIPA bias is implicitly included in the 3D effects discussed in section 4.4.*

References:
Evans, K.F., A. Marshak, and T. Várnai, 2008: The potential for improved cloud optical depth retrievals from the multiple directions of MISR. J. Atmos. Sci., 65, 3179-3196.
Frame, J. W., J. L. Petters, P. M. Markowski, and J. Y. Harrington, 2009: An application of the tilted independent pixel approximation to cumulonimbus environments. Atmos. Res., 91, 127–136.
Várnai, T., and R. Davies, 1999: Effects of cloud heterogeneities on shortwave radiation: Comparison of cloud-top variability and internal heterogeneity. J. Atmos. Sci., 56, 4206–4224.
Wapler, K., and B. Mayer, 2008: A fast three-dimensional approximation for the calculation of surface irradiance in large-eddy simulation models. J. Appl. Meteor. Climatol., 47, 3061–3071.

***13) p10,l8: Why does "non-aggregated" have coarser resolution? Isn't it just the opposite?***
Here the "non-aggregated" leads to confusion. What we mean is that 3D reflectances are aggregated while 1D reflectance are not because they are computed from the aggregated optical thickness..
We simply remover "non-aggregated" before 1D reflectance to avoid the confusion

***Summarizing, the factual problems seem to lie in a rather superficial interpretation of the findings, and they could benefit from discussion with co-author Tamas Varnai and other experts in the field. The problems are compounded by many language errors, and I advise to run a spell and grammar check, and further to go through punctuation and semantic/idiomatic use of words. Such issues are not within the purview of manuscript reviewers. The time spent on this review is somewhat out of proportion with the current overall level of maturity of the manuscript.***

We change the structure of the paper as follow:
- We now present the total bias in section 4.1
- The PPHB is presented in section 4.2
- The THEAB is presented in section 4.3
- And the 3D effects are presented in section 4.4.
- We add Fig. 4 and 5 to illustrate the THEAB and 3D effects, respectively.
- The analyze of Fig. 7, Fig. 8, Fig. 9, Fig. 10 and Fig. 12 has been improved and made more clear for the reader.

We apologize for the grammar and semantic errors. We have now check the all manuscript and hopefully corrected them.

*Examples in no particular order:*

*Figure 8 caption: In 1D (top panel) [missing comma] the right column can be highlighted [should be "illuminated" - semantic error] by the photon coming from the Sun [missing comma] while in 3D, a [an optically] thick neighbor region intercept [intercepts?] first the photon [first intercepts ?] and scatted [scattered] it back to space. Aside from the errors, this statement is also hard to understand. Also, what is the difference between "intercept" and "scatter"? Physically accurate would be "scatter" or "attenuate".*

Thank you, we have made the necessary change and rephrase the sentence: "*In 1D (top panel), the right column can be illuminated by the photon coming from the Sun, while in 3D (bottom panel), an optically thick neighbor region scatters first the photon, increasing the reflectance of the thick region, but reducing the reflectance of the thin region.*"

*"the variety of voxel extinctions from a line of sight to another can be quite similar". (In this case, it's unclear what this means - perhaps that the extinction along the line of sight varies little from one tilted column to the next?)*
Yes this is exactly what we mean and we rephrase it like you suggest.

*"This is because of the THEAB which is a positive bias, stronger at high resolutions and large view angles." Not a sentence.*
An "is" was missing there. Thank you.

*"are more highlighted from the side" - this should be "illuminated" throughout the manuscript, unless the intention was to say "highlight", but that doesn't seem to make sense.*
We agreed that in this context "highlighted" should be replaced by "illuminated". We apologize for the semantic error and make the necessary correction through the manuscript.
*"depriving neighbor cloudy columns from [of] incoming photons" – aside from the wrong preposition, using "deprive" seems inappropriate for an inanimate object.*
We removed the "of" and changed "depriving" in "blocking"
*"an important factor that constrains the impact of these assumptions" - "determines" instead of "constrain"?*
Yes, thank you.

*"To compare reflectances issue from a 3D radiative transfer: : :" use of "issue" is unclear [as noun or verb]*
We have removed "issue"
*"conversevely" [sic] - several such typos that a spell checker would pick up*
This has been corrected
*"have an almost nil effect" - wrong semantic context for "nil"*
"nil" has been changed by "no".

*"which becomes almost null" - zero? idiomatic/semantic error*
Corrected, thank you.

*"since less different cloudy columns are crossed" - it should be "fewer" instead of "less"*
Corrected, thank you.

*"the PPHB increases as the spatial resolution increases" (the intended phrasing was probably: "the PPHB increases as the spatial resolution decreases (pixel size/aggregation level increases)".*

Yes, we agreed, this is now corrected, thank you.

*"the absolute 3D effects are slightly smaller and follow the same decreasing with coarsening spatial resolution" - "follow the same decreasing with coarsening" does not seem to work. Perhaps "also decreases with coarser resolution"?*
Corrected, thank you.

*"fallstreaks or not" - "whether fallstreaks are included or excluded"?*
Corrected, thank you.

*W.m-2: What is the meaning of the dot - found throughout the manuscript?*
We are not sure to understand what referee #1 asks but instead of the dot we now

*"can be extrapolated to other cirrus clouds" - "generalized" instead of "extrapolated"?*
Corrected, thank you.

*"spatial resolutions considered here are ranged from : : :" - spatial resolutions considered here range from : : :*
Corrected, thank you.

*"most of the figures shown [showed] the"*
Corrected, thank you.

*"THEAB and PPHB is [are] complicated"*
Corrected, thank you.

*"view zenith angle" - "viewing" instead?*
Corrected through the manuscript, thank you.

*"because of no THEAB" - because THEAB is turned off [or some qualifier instead of a "no"]*
Corrected, thank you.

*p2,l22-23: What is the difference between "information content" and "retrieval methods" in this case? They are two different categories.*
This is indeed confusing, we rephrase it as:
"…but the number of retrievable cloud parameters is limited by the information content of the radiative measurements."
*p4,l2: "LES domain" - was LES introduced before?*
No it was not but we removed "LES" in the updated version of this sentence.

References:
Fauchez, T., S. Platnick, O. Sourdeval, K. Meyer, C. Cornet, Z. Zhang and F. Szczap: *Cirrus Heterogeneity Effects on Cloud Optical Properties Retrieved with an Optimal Estimation Method from MODIS VIS to TIR Channels.*, AIP Conf. Proc. 1810, 2017.

---

## Author Response (AR1)

**Scale dependence of cirrus heterogeneity effects. Part II: MODIS \*NIR and SWIR channels**

Thomas Fauchez1,2, Steven Platnick2, Tamás Várnai3,2, Kerry Meyer2, Céline Cornet4, and Frédéric Szczap5 1Universities Space Research Association (USRA), Columbia, MD, USA 2NASA Goddard Space Flight Center, Greenbelt, MD, USA 3University 
[revised manuscript text omitted]

**15 4.1 Horizontal heterogeneity and 3D effects Overall differences between 3D and 1D reflectances**

20

In nature, radiative transfer occurs in 3D not in 1D. Therefore, in addition to the PPHB, 3D radiative effects influence the spectral reflectance of a given pixel due to its radiative connection betweento its neighbors. These 3D effects includes various effects such as the HRT between cloudy columns or side illumination and shadowing effect for oblique Sun illumination. To compare reflectances issue from a 3D radiative transfer through a heterogeneous pixel with reflectances from the 1D homogeneous pixel assumption, we estimated the arithmetic mean difference between aggregated 3D at x km and non-aggregated (coarser-resolution) 1D

reflectances of the mean optical thickness at x km, with respect to the 3D aggregated reflectance in percentage, as follows:

$$\overline{\Delta R}(\overline{3D} - 1D) \ (\%) = \frac{100}{\overline{R^{3Dxkm}}} \times \left[\sum_{i=1}^{N} (\overline{R^{3D}}^{xkm} - R^{1Dxkm})\right]/N,\tag{1}$$

with where  $\overline{R^{3D}}^{xkm}$  is the averaged of 3D radiances reflectances computed at 50 m resolution,  $R^{1Dxkm}$  is the 1D radiances reflectances computed at for the optical thickness averaged over  $x \ km$  and N is the number of pixels at the spatial resolution

25  $x \, km$ . Note that because the PPHB is already included in Eq. 3, the comparison here shows the total bias including how the nonlinearity of the relationship between reflectance and optical thickness, combined with and the 3D radiative effects and solar geometries, affects TOA reflectances for a given view angle and spatial resolution.

Some effects such as the HRT may have almost nil effectno impact on average reflectances but locally, at the pixel scale, 30 they may have large positive and negative magnitudeseffects. We therefore estimate the mean absolute magnitude of the total effect by calculating the absolute mean difference between aggregated 3D and non-aggregated 1D reflectances, relative to the 3D aggregated reflectance in percentage, as follows:

$$\overline{|\Delta R}(\overline{3D} - 1D)| \ (\%) = \frac{100}{\overline{R^{3Dxkm}}} \times [\sum_{i=1}^{N} (\overline{|R^{3D}}^{xkm} - R^{1Dxkm}|)]/N,$$
(2)

Figure 8 shows ΔR(3D - 1D) (panels (a), (b) and (c)) and |ΔR(3D - 1D)| (panels (d), (e) and (f)) at 0.86 μm as a
function of the spatial resolution (ranging from 50 m to 10 km), for various viewing and solar angles. Firstef all, we see that ΔR(3D - 1D) is on-average negative for most of the spatial resolutions, viewing and solar angles and is larger than the 3% MODIS reflectance measurement uncertainty., mainly due to the HRT (see Fig. 6) and PPHB for optical thicknesses larger than 5 (see Fig. 3)Howevever, at Θv = 60° we see that ΔR(3D - 1D) changes sign. This is because the THEAB, which is a positive bias, is stronger at high resolutions and large view angles (see Fig-13). Indeed, as previously stated, in 3D RT the HRT acts mostly by moving photons from thick to thin areas leading to an increase of reflectances for small optical thicknesses and a decrease of reflectances for large optical thicknesses in comparison to 1D RT. Furthermore, [ΔR(3D - 1D)] is, on average, decreasing with Θs (except at Θv = 60°; Φv = 180°), because the PPHB is stronger (at coarser resolutions, cf. Fig-11 (d), (e) and (f)) and because the HRT from thick to thin areas is mitigated by the side illumination effect (at higher spatial resolutions). In 3D, the side illumination effect leads many photons to be first intercepted by thick regions, without reaching thin regions, contrary to cases with an overhead Sun. In turn, this leads to a larger reflectance in thick regions and to a smaller reflectance in thin regions, but on average, 3D reflectances at Θs = 60° are closer to 1D reflectances than for an interaction of the side illumination of the side at larger spatial resolutions).

- 15 overhead Sun (see Fig. 7 for illustration For nadir view,  $\overline{\Delta R}(\overline{3D} 1D)$  and especially  $|\overline{\Delta R}(\overline{3D} 1D)|$  tend to be the smallest for nadir view because of no THEAB see explanation in section 4.3, and they are almost constant over the wide ranges of spatial resolutions and  $\Theta_s$ . For oblique views, the larger the viewing zenith view angle  $\Theta_v$ , the larger is  $\overline{\Delta R}(\overline{3D} - 1D)$  and  $|\overline{\Delta R}(\overline{3D} - 1D)|$  (see THEAB in section 4.3), except for  $\Phi_v = 45^\circ$ . This view is directly parallel to the fallstreaks of the cirrus, where the variability along the line of sight is the smallest (see Fig. 1 (b) and (e)). We can also see that due to the THEAB,  $\overline{\Delta R}(\overline{3D} - 1D)$  is positive for  $\Theta_v = 60^\circ$
- 20 for several  $\Phi_v$  for the highestfinest spatial resolutions (see section 4.3 on the THEAB). Indeed by comparing these results with those of Fig. 13, we can see that, for spatial resolutions below 1 km, the THEAB is the dominant effect for large solar zenith angles. The absolute THEAB effect  $\overline{|\Delta R}(1Do.e - 1D)|$  is even larger than the total effect  $\overline{|\Delta R}(\overline{3D} - 1D)|$  which is reduced by the radiative smoothing.

Figure 9 is the same as Fig. 8 but for 2.13 μm reflectances. shows ΔR(3D - 1D) (panels (a), (b) and (c)) and [ΔR(3D - 1D)] (panels 25 (d), (e) and (f)) at 2.13 μm as a function of the solar zenith angle Θs for and for various viewing angles. Comparing with Fig. 8 for 0.86 μm reflectances We can see that the amplitude of ΔR(3D - 1D) and [ΔR(3D - 1D)] are smaller at 2.13 μm for low solar zenith angles (Θs = 0 and 30°) very similar. 
[revised manuscript text omitted]

---

## Author Response (AR2)

**REPLY TO ANONYMOUS REFEREE #2**

Dear authors,
I read the response to the review #2 and appreciate that the comments were addressed thoroughly. Thank you for the clarification regarding THEA and TIPA.

One important comment though regarding the transport of photons in inhomogeneous atmospheres: The net transport of photons does not necessarily occur from optically thick to thin regions because the spatial context and scale over which the transport occurs needs to be considered. In some cases, such as the figure that you included in the response, the transport does indeed occur from "thick" to "thin", but imagine a case where a pixel has an optically thicker neighbor right next to it, but then an optically thinner pixel that follows "next door". Which direction does the transport go then? An extreme example for this scenario is an isolated cumulus cell, which obviously has internal variability in terms of the optical thickness, so following the assertion that "bright pixels get darker", and "dark pixels get brighter" that one might make after looking at the stratocumulus figure you included, one would think that this would also happen in the isolated cloud. Instead, almost every pixel in the cloud will get darker because the cloud scatters radiation into the clear-sky regions surrounding it, regardless of the immediate neighbors, which may have a larger optical thickness. This is what I mean by "spatial context" and "spatial scale".

I hope this is making sense; I would like to emphasize that my statement does mean the net effect of the direction of photon transport. There is no set-in-stone correlation between the gradient in optical thickness and the direction and magnitude of net photon transport. This is not just a theoretical assertion, but has been published for observed clouds.

I believe that the manuscript is ready to be published, but I'd like to avoid a statement regarding the direction of net photon transport that is not generally valid.

We would like to thanks one more time referee #2 for his careful reading of the manuscript. Referee #2 comments and suggestions have significantly improved the substance and the form of the manuscript. Concerning the particular point of this revision, please find our reply below:

We believe that technically, our assertion that "bright pixels get darker", and "dark pixels get brighter" still holds for the isolated cloud example the referee #2 described. As referee #2 pointed out, the bright pixels (that is, pixels containing the isolated cloud) get darker. One could argue that the surrounding darker (cloud-free) pixels get brighter by the radiation scattered toward them from the cloud sides—that is, through the 3D process mentioned in some papers dealing with aerosol remote sensing (e.g., Wen, G., A. Marshak, and R. F. Cahalan. 2008. "Importance of molecular Rayleigh scattering in the enhancement of clear sky radiance in the vicinity of boundary layer cumulus clouds." J. Geophys. Res.. 113, doi: 10.1029/2008JD010592).

On the other hand, we agree that, in some particular situations, small-scale variations in density can lead to that the net radiation goes from thin to thick regions. Therefore, to address this situation we add the following sentence in the manuscript after the description of the net radiation flux, page 8 around line 5:

[revised manuscript text omitted]